# Seville, a Cultured and Influential Court: The Palace of Ibn 'Abbād

Ignacio González Cavero 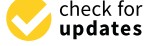

Departamento de Historia y Teoría del Arte, Facultad de Filosofía y Letras, Universidad Autónoma de Madrid, 28049 Madrid, Spain; ignacio.gonzalez@uam.es

**Abstract:** The palace of Ibn 'Abbād has long been the subject of study by numerous specialists, amongst whom its possible location has been considered to be the current site of the Reales Alcázares of Seville (Royal Palace and Fortress of Seville). The results derived from the archaeological interventions carried out in several sectors of this palatine complex have led us to undertake a review of all those studies that have dealt with this issue, to which we must add the rigorous analysis of the documentary sources to which we have access. In this sense, we intend to show the importance that this palace had between the 11th and 13th centuries, as well as its relationship with the rest of the official and residential spaces that make up the Alcazar of Seville. This made it necessary to return to the initial discussion regarding its location and to answer some questions that remain open to this day.

**Keywords:** Seville; Taifa; 11th century; alcazar (palace or fortress); palace of Ibn 'Abbād; *Qaṣr al-Mubārak*; al-Mu'taḍid; al-Mu'tamid

## 1. Introduction

The palace of al-Mu'tamid in Seville, referred to by some authors in written sources as *Qaṣr al-Mubārak* (Blessed Palace) or the palace of Ibn 'Abbād, has long been the subject of study by numerous specialists. Several documentary and material testimonies have led historiographers to question its possible location in the capital of the 'Abbādī kingdom—a reality that has led to different interpretations in this regard[1]. The importance that the aforementioned palace attained in the second half of the 11th century, along with its occupational continuity during the subsequent years, have contributed to even deeper investigation of this aspect, as well as of the conditions under which it could have reached our time. As a consequence of this prevalence, we must not forget the research carried out on the Almohad presence in the Iberian Peninsula from the middle of the 12th century and the choice of Seville as the Andalusian capital, since this knowledge and understanding is fundamental for the subject under study.

As we know, José Guerrero Lovillo (1974, pp. 97–109) identified the *Qaṣr al-Mubārak* with the palace of Pedro I (Peter of Castile) in the monumental complex of the Royal Palace of Seville, which was redecorated in the 14th century and whose extension may have reached as far as the old Casa de Contratación (House of Trade). In the opinion of some authors (Manzano Martos 1995a, p. 115; Vigil-Escalera Pacheco 1999, p. 39), the latter is where the Banū Mardanīš settled after their adherence to Almohad doctrine (*tawḥīd*) in the year 1172. However, studies derived from the archaeological interventions carried out since 1997 in some of the sectors of the fortress where the Castilian monarch undertook his constructive enterprise have questioned this theory, based on the fact that the existence of previous Almohad-era structures—erected in an earlier Taifa neighborhood—makes the survival of an 11th century palace impossible in this entire area (Tabales Rodríguez 2000, pp. 29–40; 2001a, II. 224–241; 2001b, pp. 209–12; 2001c, pp. 22–34; 2003, pp. 8–19; 2005a, II. 51–76; 2005b, pp. 8–29; 2006, pp. 6–29)[2].

As a consequence of these results, in recent years, the possibility of the *Qaṣr al-Mubārak* being located inside the area corresponding to Enclosures I and II of the Reales Alcázares of Seville (Figure 1) (González Cavero 2011) has already been advanced—that is, in the walled space that, in a generalized way, was related to the *Dār al-Imāra* (Casa del Emirato/House of Viziers) of the early 10th century referred to by the geographer al-Bakrī (d. 1094) (Al-Bakrī 1982, p. 33)[3], as well as its later extension to the south. In contrast to traditional historiography, during the first years of the present century, Miguel Ángel Tabales Rodríguez (2003, p. 13; 2005a, II. 62–63; 2005b, pp. 9–13; 2006, pp. 16–20) dated this entire palatine complex as an ex novo construction of the 11th century[4]. In fact, the recent finding of a palace in houses no. 7–8 of the Patio de Banderas (Courtyard of the Flags) has allowed specialists (Tabales Rodríguez and Vargas Lorenzo 2014, pp. 9–33; 2017, pp. 210–15; Vargas Lorenzo 2019, pp. 1–40; 2020a, pp. 209–58) to endorse this proposal, placing its chronology between the end of the 11th century and the beginning of the 12th century.

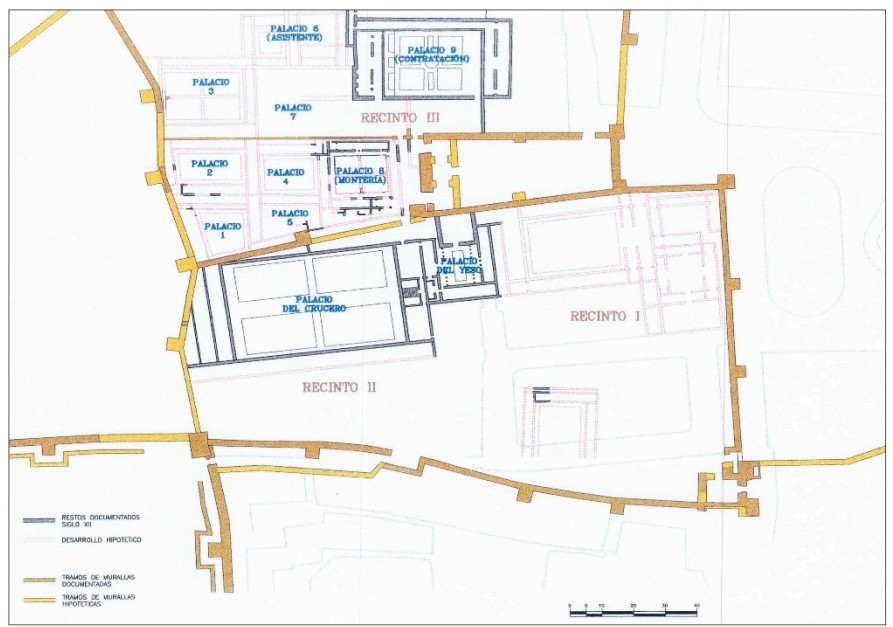

**Figure 1.** Plan of the Alcazar of Seville in the Almohad period (Tabales Rodríguez 2005b, Figure 10).

However, despite the advances made in the last two decades, we will have to wait for future research to continue to shed some more light on this matter. Nevertheless, and based on the documentary data to which we have access, we consider it to be of great importance to reflect, first of all, on the different news that alludes to the palace of al-Muʿtamid of Seville—as well as on the different terms used to refer to it—with the aim of subsequently completing its analysis, taking into account the studies carried out and the material evidence that has become available to us.

## 2. The Palace of al-Muʿtamid in Seville in the Arabic Documentation

Before focusing on the main corpus of this section, we would like to highlight how, over the years, the different authors who have referred to this palace in their works have done so using different expressions. This is an aspect that should not be overlooked, in spite of the literary connotations that characterize the Arabic written documentation and context in which the texts were conceived (Martos Quesada 2022), since it will allow us to take a step further in our knowledge, and it is also fundamental to consider the context in which it is framed. Hence, we take this reality as a starting point, following the chronology of the different historical and political periods that followed one another in al-Andalus for a better understanding of this whole scenario.

In this sense, it is precisely in the genre of 11th century poetry that we find the first documentary reference to the *Qaṣr al-Mubārak*. The Cordovan poet Abū-l-Walīd b. Zaydūn

(1003–1071), secretary and vizier of the 'Abbādī court between the years 1049/1050 and 1069/1070, is noted for his poetic production (*Dīwān*), collected by later authors such as Ibn Jāqān (d. 1134) and Ibn Bassām (d. 1147)[5]. Among his literary compositions he extols the "palace of al-Mubārak" and highlights one of its sumptuous formal halls, known by the name of *al-Ṯurayya* (The Pleiades), implying in his verses the close relationship between the latter and al-Muʿtamid of Seville (1069–1091):

"As for the *Ṯurayya*, it resembles the Pleiades
for its high position, its usefulness and beauty.
If he does not receive your visit, al-Muʿtamid, desires it so much,
that he will be at your side through imagination.
He will drink at your side every day to enjoy a quiet joy:
Extend your visit so that he may be happy.
It would seem to thee that the palace of *al-Mubārak* is like the
cheek of a girl in the center of which *al-Ṯurayya* is
like a mole.
Go around with a wine glass of the most perfect perfume
and of the color of pure gold.
It is a palace that rejoices the eyes with a construction of spacious rooms
and if it could, it would be proud of its beauty"[6].
[Original translation into Spanish:
En cuanto a la *Ṯurayya* se asemeja a las Pléyades
por su alta posición, su utilidad y belleza.
Si no recibe tu visita, al-Muʿtamid, lo desea tanto
que iría a tu lado con la imaginación.
Va a beber a tu lado todos los días para gozar de una
alegría tranquila: Prolonga tu visita para que sea feliz.
Te parecería que el palacio de *al-Mubārak* es como la
mejilla de una muchacha en cuyo centro *al-Ṯurayya* es
como un lunar.
Haz circular allí una copa de vino del perfume más
perfecto y del color del oro puro.
Es un palacio que regocija a los ojos con una construcción
de amplias dependencias y si ella pudiese, se enorgullecería
por su belleza].

Even King 'Abbādī himself remembered this palace, among others, during the years of his exile in Agmāt (1091–1095) by the Almoravids, as he left on record in the following verses, and which Ibn Jāqān collects in his *Qalāid al-ʿiqyān* (Golden Necklaces):

"Cry *al-Mubārak* for the memory of Ibn 'Abbād,
cries for the memory of lions and gazelles.
He cries for his *Ṯurayya* because he is no longer covered by its stars
which resemble the sunset of the Pleiades when it rains [ . . . ]"[7].

[Original translation into Spanish:
Llora *al-Mubārak* por el recuerdo de Ibn 'Abbād,
llora por el recuerdo de los leones y las gacelas.
Llora su *Ṯurayya* porque ya no le cubren sus estrellas
que se parecen al ocaso de las Pléyades cuando llueve [ . . . ]].

Moreover, and although his name is not mentioned as in the previous poems, some scholars suggest that the palace referred to by the Sicilian poet Ibn Ḥamdīs (d. 1133) in one of his poetic compositions is none other than the *Qaṣr al-Mubārak* (Rubiera [1981] 1988, pp. 136–37; Pérès 1983, p. 144; Gómez García 2004, p. 269). Thus, we can corroborate it when he compares said construction with the *Iwān* of Cosroes, basing this statement



on the link established between the two—on this occasion, explicitly—by the poet Abū Ŷaʿfar b. Aḥmad (11th–12th centuries) in his *Risāla* (*Epistle*), collected by Ibn Bassām (1979, pp. 759–66). In that composition, the author personifies two great palaces of Seville—the *Qaṣr al-Mubārak* and the *Qaṣr al-Mukarram*—establishing a dialogue between them in which the former says "If the Iwān of Cosroes were my contemporary, I would still have, despite its existence, power and fame" (Lledó Carrascosa 1986, p. 197).

Returning once again to Ibn Ḥamdīs's poem, his verses equate the characteristics of its construction with the qualities of the prince by pointing out how "[ . . . ] from his chest [the builders] have taken their breadth, from his light the brightness; from his fame, the wide distribution and from his wisdom, the foundations" (Rubiera [1981] 1988, p. 136; Pérès 1983, p. 144). As we can see, he does not specify at any time to whom he is referring when mentioning the "prince". In our opinion, it could be al-Muʿtamid himself since, prior to the year 1078, Ibn Ḥamdīs was in Sicily, and it would not be precisely until then when he appeared in the environment of the poetic court of al-Muʿtamid.

Therefore, at this time, he could have written this poem in praise for the current monarch and with the aim of becoming part of his cultural circle (González Cavero 2013, II. 58–59). Even the Moroccan compiler ʿAbd al-Wāḥid al-Marrākušī (1185-m. after 1224) extols the virtues of the latter [8], providing support for our proposal. In view of this, and if it is really the *Qaṣr al-Mubārak*, we can deduce through all of the above not only the importance that this palace attained in the 11th century as the nerve center of power and of the Taifa of Seville, but also the existence of an ex novo construction—albeit with certain nuances, as explained below.

That relevance of which we speak is evidenced in the *Risāla* (*Epistle*) of Abū Ŷaʿfar b. Aḥmad (Lledó Carrascosa 1986, pp. 191–200), to which we alluded earlier. In it, the denier author—exiled from the court of al-Muʿtamid in the year 1076—clearly distinguishes between two palaces: the *Qaṣr al-Mukarram* (Worshiped Palace) and the *Qaṣr al-Mubārak* (Blessed Palace), going so far as to compare the latter to the Kaʿba after having become "the goal of travelers, the center of pilgrims" (ibid., p. 197). Such an expression should not be strange to us, since we know that Seville had by then become the intellectual center of greatest importance in the entire peninsula, to which a multitude of literates and sages flocked to form part of the ʿAbbādī cultural environment (Lirola Delgado 2013, pp. 107–10).

Sharing the opinion of Lledó Carrascosa, everything seems to indicate that the *Qaṣr al-Mukarram* was the older of the two, with al-Muʿtamid himself being in charge of renovating it and, at the same time, reviving the *Qaṣr al-Mubārak*, thereby giving the latter the prominence that it deserved. Does this mean that said palace may have existed prior to the reign of the last sovereign of the Sevillian Taifa[9]? Apart from the latter, this relationship between this palatine construction and the aforementioned monarch is something that is constant in the texts, and it is also mentioned in the documentary sources as "al-Muʿtamid's palace" or "Ibn ʿAbbād's palace". Thus, we can verify it on the occasion of the assassination of al-Muʿtamid's friend, poet, and prime minister Ibn ʿAmmār (d. 1086) at the hands of the Sevillian king himself—an event recorded by ʿAbd al-Wāḥid al-Marrākušī in his work. Despite the later date of this work relative to the events that it narrates, it is significant to note how the author clearly identifies the one mentioned in the 11th century as *Qaṣr al-Mubārak* with the "Palace of al-Muʿtamid", which allows us to endorse this link with the figure of this monarch:

> "[Ibn ʿAmmār] He was placed in a high room over the door of the Palace of al-Muʿtamid, known as the blessed Palace -al-Qaṣr al-mubārak-, which subsists until this our time [ . . . ] but nothing bent al-Muʿtamid and he wounded him with the axe he had in his hand, not ceasing to wound him until he lay dead. He withdrew al-Muʿtamid and had him washed and shrouded; he made the prayers for him and buried him in the blessed Al-Mubārak" (ʿAbd al-Wāḥid al-Marrākušī 1955, pp. 98 and 100)[10].

Regarding this same episode, some Arab authors closer in time to the events—such as the already-cited Ibn Jāqān and Ibn Bassām, and even (somewhat later than ʿAbd al-

Wāḥid al-Marrākušī) Ibn Jallikān (1211–1282)—also cite this palace with the name of *Qaṣr al-Mubārak*, although they differ from the Moroccan compiler about the burial place of Ibn ʿAmmār:

> "[Al-Muʿtamid] hit him with the axe, and ordered him to be finished off and taken away and he was hidden with his chains, outside the gate of the palace (al-qaṣar) al-Mubārak, known in Seville as Bāb al-Najal"[11].

Apart from the latter, this association of the ʿAbbādī monarch with the Seville Palace is not exclusive to this particular case, as Ibn Jallikān (1868, IV. 457) himself clearly highlights in his biographical dictionary *Kitāb wafayāt al-aʿyān* (*Book of illustrious men*). The Iraqi author records that the Almoravid emir Yūsuf b. Tāšufīn (1062–1106) was amazed by the palaces of al-Muʿtaḍid (1042–1069) and his son and successor al-Muʿtamid when he entered the Sevillian capital after the battle of *al-Zallāqa* (Sagrajas) in 1086, settling in one of them. We do not know in which one he could have stayed, but it would not be surprising if he did so in the much-acclaimed *Qaṣr al-Mubārak*. However, this also allows us to verify the existence of several palaces in Seville in the 11th century, some of which al-Muʿtamid would refer to during his exile in Agmāt, as we have seen.

This event is also mentioned by al-Ḥimyarī (13th–14th centuries) and al-Maqqarī (d. 1632), referring also to al-Muʿtamid's Palace—this time as Ibn ʿAbbād's palace—when, around the years 1082–1083, Alfonso VI of León (1065–1109) sent his troops to Seville to besiege the aforementioned palace (Al-Maqqarī [1843] 2002, II. 289–292)[12]. In fact, al-Ḥimyarī adds that it was located in the vicinity of the Guadalquivir, which allows us to know its approximate location, as we can interpret from the following fragment:

> "When Alfonso received the news of what Ibn ʿAbbād had done, he swore to his great Gods, that he would go as far as Seville to attack him, and that he would besiege him in his own palace. He put two armies on a warpath; he entrusted the command of one of them to one of the mangy men who served as his generals, and ordered him to set out [ . . . ] He met him at Triana, where they would carry out their union. For his part, Alfonso, at the head of a second army, quite numerous, took a different path from the one his general was to follow. They both plundered the Muslim territory, sowing ruin and desolation, then they held their meeting at the place set on the edge of the Guadalquivir, in front of the palace of Ibn ʿAbbād" (Al-Ḥimyarī 1938, pp. 105–6 [trans.]; 1963, pp. 175–76)[13].

At this point, we can see how the written documentation tells us about a palace in the context of the 11th century that belonged to the last king of the Sevillian Taifa, by using different expressions such as "Qaṣr al-Mubārak"—mainly in the poetic genre—"palace of al-Muʿtamid", and "palace of Ibn ʿAbbād", the latter two being the most used by Arab authors from the 12th century onwards. Therefore, based on this reality and the analysis carried out, we can confirm that it is the same construction. However, all of this contrasts with the absence of any references to the aforementioned palace during the years that al-Andalus was part of the vast Almoravid Empire. It is in the Almohad era that we again find numerous documentary references to this construction, whose authors continue to allude to it in different ways and, moreover, with different functions. For this reason, we consider it essential to compare not only the different versions of the texts that we have, but also to resort to their Arabic editions.

In the first place, the Moroccan compiler Ibn ʿIḏārī al-Marrākušī (d. after 1313) already confirms in his *Al-bayān al-mugrib fī ijtiṣār ajbār mulūk al-Andalus wa-l-Magrib* (*The surprising exposition in the summary of the news of the kings of al-Andalus and the Maghreb*) the existence of the palace that we are addressing at that time, referring to it by the name of "palace of Ibn ʿAbbād" as the place where the sheikhs Abū Yaḥyà b. al-Ŷabr and Abū Isḥāq Barrāz settled after their entry into Seville in the mid-12th century:

> "[the Almohad armies] (were installed) inside Seville, so that they would be close to the palace (qaṣr) of Ibn ʿAbbād where the (two) Almohad sheikhs Abū Yaḥyà b. al-Ŷabr and Abū Isḥāq Barrāz, the head of the Government (majzan) with

the High Command (al-amr al-ʾalī) resided, and thus they would be (all) the Almohads together with one another"[14].

This aspect leads us to consider a double reality: on the one hand, we can see how the old palace of al-Muʾtamid in Seville was still standing in the middle of the 12th century and, on the other hand, its evident occupational continuity as the most important political–administrative nucleus of the city linked to the court, and which the Almohad sheikhs chose from the beginning as their residence. This demonstrates not only the prominence that the 11th century palace maintained during the following century, but also the clear intention on the part of the Almohads to assume or, rather, to legitimize, their power through what was and continued to be a courtly reference.

Its memory was preserved throughout the period of Almohad domination in the Iberian Peninsula. Hence, it is the chronicler of the Almohad court Ibn Ṣāḥib al-Salā (d. after 1198) who provides us with the most data, being also aware of the contemporaneity of the events that he narrates and his link with the North African state apparatus. It is in his chronicle *Al-mann bi-l-imāma* (*The divine gift of the imamate*) where we not only continue to find numerous references to the palace of al-Muʾtamid as the "palace of Ibn ʿAbbād", but also find new expressions derived from the versions that we have, and which we consider important to take into account for a later analysis.

As for its location, and as we can interpret from the aforementioned work, Ibn Ṣāḥib al-Salā coincides in locating the palace of Ibn ʿAbbād in the same place that seems to emerge from the events reported by al-Ḥimyarī and al-Maqqarī on the occasion of the siege undertaken by Alfonso VI. We can see this when the author of Beja narrates how, a few years earlier—specifically, in 1161–1162—the rebel Ibn Abī Ŷaʿfar was crucified in the sandbank, near the palace of the last king of the Taifa of Seville:

> "Upon taking it [Carmona], the Levantine caid Ibn Abī Ŷaʿfar was caught and was taken in chains to the prison of Seville, and remained there until the obeyed order came that he be crucified. This was done in the sandbank (Rambla), under the citadel or fortress of Ibn ʿAbd in Seville, and the rebellion in Carmona was ended, as I expounded it in History" (Ibn Ṣāḥib al-Salā 1969, p. 37).

As we can see, the chronicler of the Almohad court specifies more in this regard, indicating that the palace of Ibn ʿAbbād was built in the vicinity of the area then occupied by the sandbank—that is, in the area between the current course of the Guadalquivir, the Puerta de Triana (Triana's Gate), and the Torre del Oro (Golden Tower), which leads us to validate its location in the vicinity of the river. We even know that in the 11th century the river flowed closer to the primitive nucleus of the city, moving progressively towards the west and leaving space for the urban growth of Seville on this side.

The expression used by Ambrosio Huici Miranda in his translation of the *Mann bi-l-imāma* to refer to the palace of al-Muʾtamid as "castle of Ibn ʿAbd" is also significant. However, in the Arabic edition by ʿAbd al-Hādī al-Tāzī of the unique manuscript of this work of Ibn Ṣāḥib al-Salā (1964, p. 185)—preserved in the Bodleian Library (University of Oxford, England), and which the aforementioned Spanish historian and Arabist uses for his Spanish version—we can read "Qaṣr Ibn ʿAbd", or "fortress of Ibn ʿAbbād". When writing this, did Huici Miranda want to reflect the fortified character that the latter may have had?

It is in the year 1172 that we again find news of the ancient palace of al-Muʾtamid, but this time endowed with a different function from what we have seen so far. We have evidence through Ibn Ṣāḥib al-Salā that, after the death of the Levantine emir Ibn Mardanīš (1147–1172), the *Šharq al-Andalus* became part of the Almohad Empire, submitting his family to the unitary dogma (*tawḥid*) before the caliph Abū Yaʿqūb Yūsuf (1163–1184) in Seville, the Andalusian capital. According to the chronicler of Beja, once Hilāl b. Mardanīš recognized the caliph, he was received in the palace of Ibn ʿAbbād—which was noted for its magnificence and spaciousness—while his companions were hosted in houses adjacent to it:

"Hilāl b. Mardanīš bid farewell with his companions, and accommodation was procured for him and his companions. He was received in the magnificent and spacious palace of Muḥammad Ibn 'Abbād (Mutamid), amir of Seville; his companions were hosted in the adjoining houses, and beds and tapestries and meals and gifts and drinks and everything necessary was provided for them, and they understood that they were the nearest of kin and the closest of friends, and that they were cordially welcomed by the caliphal kingdom and the Imami power" (Ibn Ṣāḥib al-Salā 1969, pp. 193–94)[15].

This was also the case that same year prior to the Huete campaign[16], and even on his return to Seville after the failure of the latter:

"That day the sons of Muhammad b. Mardanīš arrived with him, with their servants and the servants of their father and brothers, as he had commanded them, with the most pompous entrance, and he received them in the palace of Ibn 'Abbād and in the houses adjoining it, and bought houses for them in Seville, from their owners, to house them." (Ibn Ṣāḥib al-Salā 1969, p. 226; 1964, p. 516).

As we can interpret from these facts, the palace of Ibn 'Abbād was also used by then as a temporary residence of the Banū Mardanīš. This apparently paradoxical reality should not be surprising to us since, in spite of the continuous confrontation that prevailed during the third quarter of the 12th century between the Almohad armies and the Mardanisid troops, the family of the Levantine rebel came to occupy a high position in the court and in the North African military apparatus, as expressed by the aforementioned chronicler of Beja. As a matter of fact, that importance to which we are referring is perfectly confirmed in the conception of this palace as a place destined for the family of Ibn Mardanīš.

At this point, we consider it appropriate to dwell on how the Almohad sheikhs Abū Yaḥyà b. al-Ŷabr and Abū Isḥāq Barrāz chose the palace of Ibn 'Abbād as the residence and seat of their government in al-Andalus in the middle of the 12th century, which must have continued to maintain, in our opinion, the same functions during the subsequent years, being used later as a place of residence for those delegations that went to Seville to take the oath of recognition (*bay'a*) to the caliph. We can see this not only at the end of the 12th century with the Banū Mardanīš, but also with Ibn al-Aḥmar of Granada (1232–1273) who, on the occasion of the renewal of the peace treaty that he maintained with Alfonso X the Wise (1252–1284), was received in the palace of Ibn 'Abbād upon his arrival in Seville:

"When Ibn al-Aḥmar arrived in Seville, he camped on its outskirts in the Red Cistern, and five hundred chosen horsemen, warlords and captains were with him. Alfonso went out to meet him and conjured him to come to him; he entered and stayed in the palace of Ibn 'Abbād and the two main leaders sons of Ašqīlūlā, Abū Muḥammad and Abū Isḥāq, entered with him [ . . . ] When Ibn al-Aḥmar entered and settled, the Christians built a high palisade in the street where he stayed, they built it at night and it remained nailed before the houses in such a way that it prevented the horses from passing" (Ibn 'Iḏārī 1954, II. 285–286).

Considering all these facts, could we think that, just as we have anticipated, the Almoravid emir Yūsuf b. Tāšufīn could have also settled in the *Qaṣr al-Mubārak* after the battle of *al-Zallāqa*? Everything seems to indicate that the "magnificent and spacious palace of Muḥammad Ibn 'Abbād"—as previously remarked by Ibn Ṣāḥib al-Salā—was endowed with different spaces with very diverse functions linked to the court, leading us to raise the following questions: On the one hand, which was the area of government or official representation where Hilāl b. Mardanīš and the rest of his companions submitted to the Almohad *tawḥid*? And, on the other hand, where did the Almohad caliph reside if the palace of Ibn 'Abbād was the place where the Banū Mardanīš and, later, Ibn al- al-Aḥmar stayed?

Regarding the first of these questions, the aforementioned Almohad chronicler refers in his work to the "old castle" as the place where Hilāl b. Mardanīš was received in April 1172 by the caliph Abū Ya'qūb Yūsuf to proceed to his recognition:

"He [Hilāl b. Mardanīš] arrived with all his brothers and with the supporters of his father, the caids and the greatest ones of the military border, at the coming of Ramadan of this year (begins April 27, 1172). The Amīr al-Mu'minīn sent to his encounter, his brother the illustrious Sayyid Abū Zakariyā' Yahyà, son of the caliph 'Abd al-Mu'min, lord of Béjaïa, and also his other brother, Abū Ībrāhīm Isma'īl [ . . . .] He entered in his company into the ancient castle at the reception of the caliph, close to the evening prayer on the day of his arrival, and then the new moon (hilāl) of Ramadān of the year 567 appeared. He greeted the caliph Abū Ya'qūb and recognized him in the presence of all the Sayyids, the illustrious Sayyid Abū Ḥafṣ and all the brothers and sheikhs of the Almohads and the Ṭālibes of the court" (Ibn Ṣāḥib al-Salā 1969, p. 193).

So did the rest of the personalities and military who arrived with him in Seville the following day. Moreover, it is precisely in this "ancient castle" that the audience room was to be found that, consequently, would form part of the protocol area, as we can interpret from the following words:

"When the first day of the month of Ramadān (April 27) dawned, the sheikhs of the Almohads and all the people, and the Ṭālibes of the capital, rose early to attend the reconnaissance of the people of Levant, already mentioned. When the caliph, Amīr al-Mu'minīn, was seated on his lofty and noble throne, the vizier Abū-'Alà Idrīs b. Ŷā'mi' came out and commanded them to enter and present themselves before him. They entered and saluted with a general greeting. Then they paid homage to him one after another, preceded by their sheik Abū 'UṮmān Sa'īd b. 'Isā, chief of the aforementioned soldiers and lord of the border, pledged themselves to obedience and entered the (Almohad) community" (ibid., p. 194).

Regarding the Arabic version (Ibn Ṣāḥib al-Salā 1964, p. 472), it names the castle as *qaṣaba al-'atiqa* (ancient fortress or *qaṣaba*), which leads us to think of a fortified construction as a place chosen for this type of ceremony and, in addition, endowed with a certain antiquity when compared to other similar buildings contemporary to these events. As we know, the Sevillian capital witnessed an important urban transformation after the Almohads entered the city—mainly in its southern sector. We have evidence through the *Bayān al-mugrib* of Ibn 'Iḏārī al-Marrākušī of the construction of a first fortress around 1150 to take in the North African armies that accompanied the sheikhs Abū Yaḥyà b. al-Ŷabr and Abū Isḥāq Barrāz on the occasion of the disputes that originated with the Andalusian population:

"[ . . . ] they agreed to build a fortress for the Almohads residing in (the neighborhood of the Cemetery) al-ŷabbāna would move to it, due to the complaints of the people against the harm they caused, after which, they determined a place (for that fortress) -the same one in which it is today-, removing its inhabitants from their houses and compensating them in the medina with other houses of the Government (majzan) . . . (The Almohads) demolished the wall of Ibn 'Abbād and with its stones they built that fortress [ . . . ]"[17].

This fortress must have been built in the vicinity of the foundational site of the Reales Alcázares of Seville (Royal Palace and Fortress of Seville), and it was extended to the west during the first years of the Almohad occupation of the city (González Cavero 2018, pp. 126–56); however, we do not have further data about it. What there is no doubt about is the frequent differentiation in the Islamic world regarding this type of construction. We refer to the existence of a palace (*qaṣr*) as the residence of the sovereign and an important area of representation inside a larger fortified space delimited by the fortress (*qaṣaba*), the latter term being understood as the wall itself that encloses all of that territory. In our opinion, we must take this aspect into account to understand the reality of Ibn 'Abbād's palace.

However, to the above, we must add that Ibn Ṣāḥib al-Salā himself seems to be referring to another fortress a year later, on the occasion of the act of congratulating the caliph Abū Ya'qūb Yūsuf after the battle against the Christians of Avila:

"Amīr al-Mu'minīn sat down and his brother the illustrious Sayyid Abū Ḥafṣ sat down with him, on Saturday the 22nd (April 8) at sunrise, for the blessed congratulatory ceremony in his palace, inside the fortress of Seville. The Almohads and the sheikhs of the Ṯālibes of the court and the alfaquis and the secretaries and the preachers were ordained, and those who came to the gate (of the palace) for the congratulatory ceremony were presented and authorized, as they entered by their literary and poetic categories [ . . . .] Then the Amīr al-Mu'minīn was recognized, following this, and all of those present kissed his blessed hand and the joy was completed with this, and this marks the beginning of the many conquests that followed" (Ibn Ṣāḥib al-Salā 1969, p. 232; 1964, pp. 524–25).

It is possibly the latter to which the chronicler of the Almohad court refers when Abū Ya'qūb Yūsuf ordered the governor of Seville Abū Dāwūd Yalūl b. Ŷaldāsan to build a strong wall "in the fortress of Seville that would pass from the beginning of its construction in front of the esplanade of Ibn Jaldūn, inside Seville, and to raise the minaret of the mosque, which would be at the junction of the wall with said mosque" (Ibn Ṣāḥib al-Salā 1969, p. 200) in the year 1184[18]. This represents a new fortress that we can identify with the one referred to by historiographers as the "inner fortress" (González Cavero 2018, pp. 217–33), and which would comprise the space between the palatine complex and the mosque (*masŷīd al-ŷami'*) (Figure 2, Enclosures VI and VII). This fortress may have been completed in 1172, once Abū Ya'qūb Yūsuf managed to consolidate his position in al-Andalus after the subjugation of the Banū Mardanīš, as part of the constructive enterprise conceived by the Almohads in this sector of the city (Ibn Ṣāḥib al-Salā 1964, p. 474; Roldán Castro 2002, p. 15).

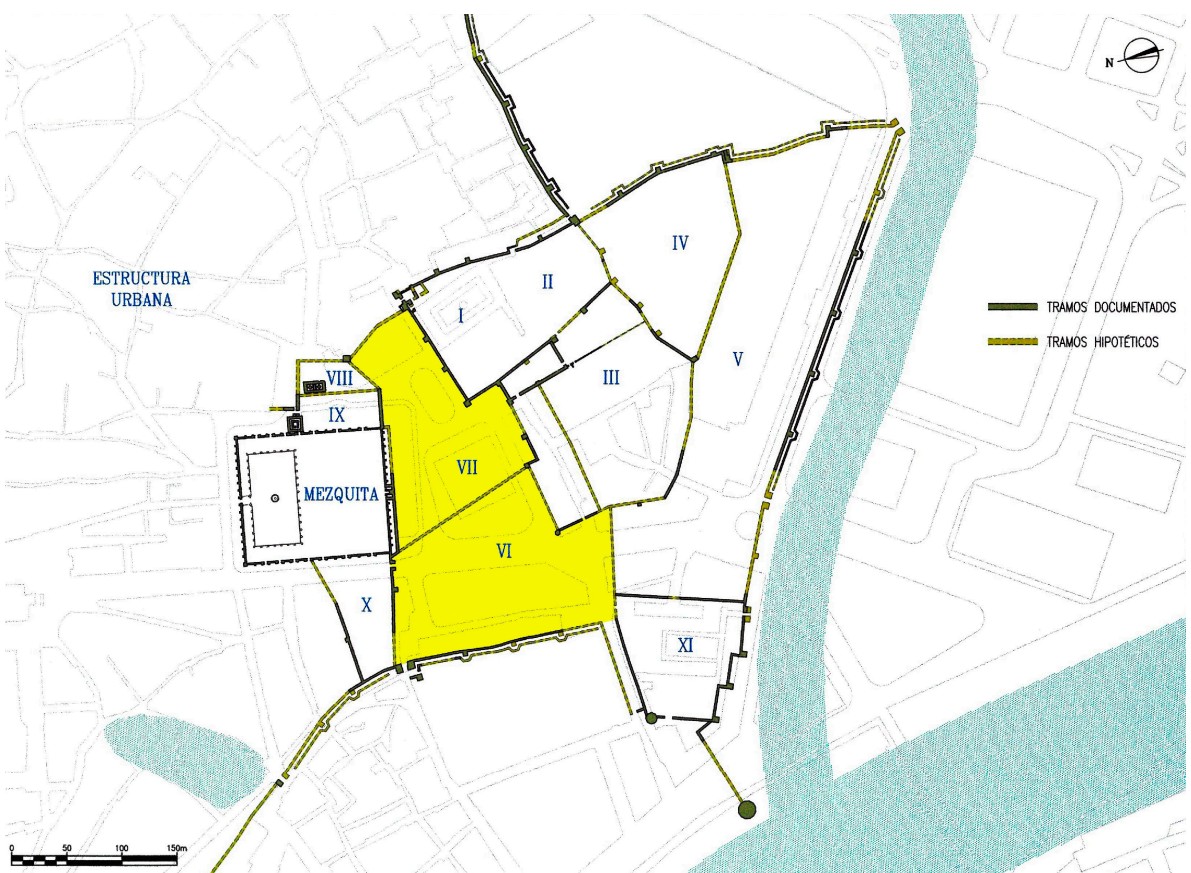

**Figure 2.** Enclosures VI and VII of the Alcazar of Seville based on the plan by Tabales Rodríguez (2010, p. 244).

If so, could we relate the *qaṣaba al-ʾatiqa* with the primitive enclosure of the Reales Alcázares of Seville (Royal Palace and Fortress of Seville) that was included in the aforementioned "inner fortress" and that has traditionally been associated with the *Dār al-Imāra* of the 10th century? For his part, Leopoldo Torres Balbás (1949, p. 30) had already identified the walls and towers of the fortress with the "old fortress" referred to at the end of the 12th century. Even Antonio Almagro Gorbea (2015a, p. 11) affirms that the oldest area would be defined by the layout of its walls built from ashlars, while the most recent ones would be built from mud or brick.

We will return to this question later but, responding to the model of this type of construction to which we referred earlier, it is significant to note how Ibn Ṣāḥib al-Salā specifies that the congratulatory session took place "in its palace, within the fortress of Seville", which leads us to think that, indeed, we could be looking at a palatine complex integrated into this new defensive area. This was suggested by Melchor Martínez Antuña (1930, pp. 69–70)[19], based on the unique manuscript of the Bodleian Library, in which he adds "the hall on the right of the palace was the one destined by the caliph Yúsuf for tribute receptions and celebration ceremonies" (ibid., note 1, p. 70). In our opinion, this hall could be the same one mentioned by Ibn ʿIḏārī al-Marrākušī in narrating the act of recognition of Hilāl b. Mardanīš[20] which, as we know, he draws from the chronicle of Ibn Ṣāḥib al-Salā. Therefore, considering all of these data, we should not rule out the possibility that the *qaṣaba al-ʾatiqa* was the same primitive palace (González Cavero 2018, p. 160), with Ibn Ṣāḥib al-Salā using this expression in a punctual way to differentiate it from those more recent Almohad constructions and, in turn, legitimize the Muʾminī dynasty in al-Andalus.

All of this allows us to return to the second question that we previously mentioned about which could have been the palace where the Almohad caliphs resided; however, we only have brief allusions that can bring us closer to its possible location. This is the case of the construction of the *sābāṮ* of the mosque in Seville that allowed "the caliph to go out through it, from the palace to this mosque to attend Friday prayers" (Ibn Ṣāḥib al-Salā 1969, p. 197; 1964, pp. 477–78). We know that the qibla wall was located in front of the northern wall of the primitive enclosure of the Reales Alcázares of Seville (Royal Palace and Fortress of Seville), providing initial evidence that the latter fulfilled that function. In addition, and when speaking of the works of the *Buḥayra* of Seville, it is likely that the chronicler of Beja is referring to this again when he states that the "Amīr al-Muʾminīn would leave his palace in Seville, on horseback, with the Almohad chiefs to inspect the work and field and to recreate himself with its pleasing view" (Ibn Ṣāḥib al-Salā 1969, p. 189; 1964, p. 466), indicating that he was in the city itself.

Apart from these data on the palace, it always comes to our attention that an author such as Ibn Ṣāḥib al-Salā—the official chronicler of the Almohad court—does not dwell on the reforms or ex novo constructions that were carried out at that time in the present Reales Alcázares of Seville (Royal Palace and Fortress of Seville)—as we can confirm from the material evidence available to us. However, this contrasts with the detailed descriptions he gives of other civil, palatine, or religious works. This is the case of the ancient Roman aqueduct that came from Alcalá de Guadaíra, the palaces of the *Buḥayra*, or the mosque.

We even have evidence from written documentation that Seville, as the Almohad Andalusian capital, became the temporary residence of the caliphs during the periods that they spent in the Iberian Peninsula at the moment of supervising the works that were being carried out in the city, or in order to prepare their military campaigns against the Christians (González Cavero 2018, pp. 100–6). Nevertheless, at no time do the authors explicitly mention the place where they settled after their arrival from the official court in Marrakech. We only know that the caliph Abū Yūsuf Yaʿqūb al-Manṣūr (1184–1199) ordered the court to be moved to the outskirts of the Sevillian capital, as specifically mentioned in the *Ḥiṣn al-Faraŷ* (ibid., pp. 250–57).

Consequently, and in spite of this lack of data, there is no doubt that the temporary residence of the caliphs had to be at the place that the Reales Alcázares of Seville (Royal Palace and Fortress of Seville) occupies today, taking into account that, during the years in

which the city became the Almohad Andalusian capital, the political–religious nucleus had already moved to its southern sector. It would even be logical to think that the residence of the caliphs was located in the vicinity of the protocol area for ceremonial reasons, being able to place it in the same *qaṣaba al-ʿatiqa* and in which the much-acclaimed *Qaṣr al-Mubārak* of the 11th century may have been found. A palace that survived between the 12th and 13th centuries with a political–administrative and, in turn, residential function, leads us to question the state it was in at that time, as there was evidence of the reuse of "the stone called «Taŷūn al-ʿādī'"», taken from the wall of the palace of Ibn ʿAbbād [*surqasr* Ibn Abd]" (Ibn Ṣāḥib al-Salā 1969, p. 201; 1964, p. 482) for the construction of the minaret of the new mosque. As we can read, the palace was surrounded by a wall of stone ashlars—a reality already recorded by the poet Abū Ŷaʿfar b. Aḥmad in his *Risāla* (*Epistle*) (Lledó Carrascosa 1986, p. 197).

## 3. Recovery and Restitution of the Primitive Palace of the Reales Alcázares of Seville (Royal Palace and Fortress of Seville)

In addition to the different documentary sources that we have about this palace and the studies carried out thereon, the findings derived from the archaeological interventions that have taken place in recent years are of great importance to try to answer a question that, for some time, has been raised in the research about the Reales Alcázares of Seville (Royal Palace and Fortress of Seville): where was the palace of Ibn ʿAbbād located? We have already seen how everything seems to point to its possible location in the old enclosure of the Sevillian palace and fortress (Enclosures I and II), destroyed or hidden by later constructions and reforms that, from the same 12th century, followed one after the other until the end of the 19th century.

Therefore, ruling out the widespread theory that the *Qaṣr al-Mubārak* was located in the place now occupied by the Palace of Pedro I and its immediate surroundings, the preventive work undertaken between 2013 and 2018 under the direction of Miguel Ángel Tabales Rodríguez for the restoration of houses no. 7 and 8 of the Patio de Banderas (Courtyard of Flags) (Enclosure I) allowed the partial recovery of the remains of a palace of some consideration and entity (Figure 3). Over the course of time, the latter was segregated and transformed—mainly in the 19th century—into the properties located in the western sector of the aforementioned courtyard, being dated between the end of the 11th century and the beginning of the 12th century. Even this (apparently hidden) importance has been evidenced throughout the centuries by its particular use, to which must be added the proportions that it possesses and the choice of its location with respect to the rest of the fortified complex.

As some authors have already pointed out, the first graphic testimony about this space that has become available to us is the plan of the Reales Alcázares of Seville (Royal Palace and Fortress of Seville) attributed to the Milanese architect Vermondo Resta (1608), in which we can appreciate the configuration of a transept garden that formed part—as it appears in writing—of the so-called "quarto del alcayde" (throne room) (Figure 4). Its imprint denotes the relevance that this reserved area must have had, in modern times, to the residence of the Alcaide (Governor) of the palace (Vargas Lorenzo 2019, p. 31; 2020a, p. 251) and the inheritor, in our opinion, of its medieval past[21]. This is supported by the written documentation itself, through which we know that King Pedro I of Castile (1350–1369) was in the "palace that they say is of iron" (López de Ayala 1779, p. 240) when he ordered the assassination of his brother, the Master of Santiago Don Fadrique, in the vicinity in 1358.

This "palace" must have been none other than the "quarto de los yessos" (gypsum room) that Vermondo Resta identifies in his floor plan and that, in the 17th century, was already differentiated and independent from the "quarto del alcayde" (throne room), possibly due to the transformations to which this sector was exposed, as we have already mentioned. However, it is significant that in the 14th century the Castilian monarch had a space of such reduced dimensions as his residence, so it would be expected that at that time

the "quarto de los yessos" (gypsum room) occupied a larger area and included the "quarto del Alcayde" (throne room). Furthermore, we know from Pedro López de Ayala that Doña María de Padilla was in the "Cuarto del Caracol" (Spiral Room)—built in the southern end of the Cuarto Real (Royal Bedroom) or del Crucero (Crossing Room)—when the murder of Don Fadrique took place (ibid., pp. 239, 241), so everything leads us to believe that this palatial area was her place of residence, leaving the "quarto de los yessos" for the exclusive use of the monarch.

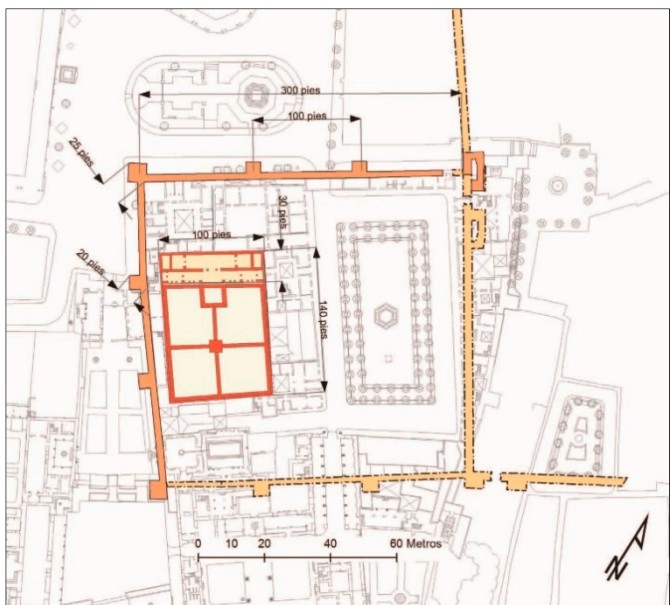

**Figure 3.** Metrological analysis of Enclosure I of the Reales Alcázares of Seville according to Tabales and Vargas (Jiménez Hernández 2015, Figure 26).

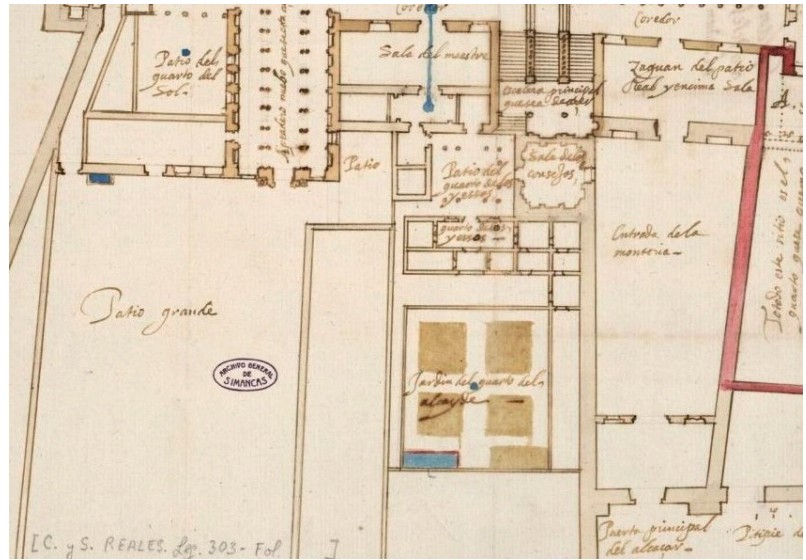

**Figure 4.** Plan of the Alcazars of Seville showing the audience room (1608). Vermondo Resta (Spain, General Archive of Simancas, MPD, 38, 134). Detail of the western sector of the Patio de Banderas.

In fact, from the plan drawn up in 1759 by the engineer Sebastián van der Borcht (Figure 5a,b), we can derive an idea of what the whole area might have been like, and in which we can already sense the layout of a palace with clear Islamic roots—despite the alterations that took place in it (Manzano Martos 1999, p. 65; Almagro Gorbea 2015a, p. 13)—that Pedro I must have occupied during his stay in Seville while his great construction

enterprise was being carried out in its vicinity. In this sense, at the end of the 20th century, Rafael Manzano Martos (1995a, p. 117; 1995b, p. 346) distinguished the remains of this construction, recreating its plan (Figure 6) and dating it to the Almohad period, as did other specialists (Torres Balbás 1949, pp. 30–31; Almagro Gorbea 2013, pp. 89–90).

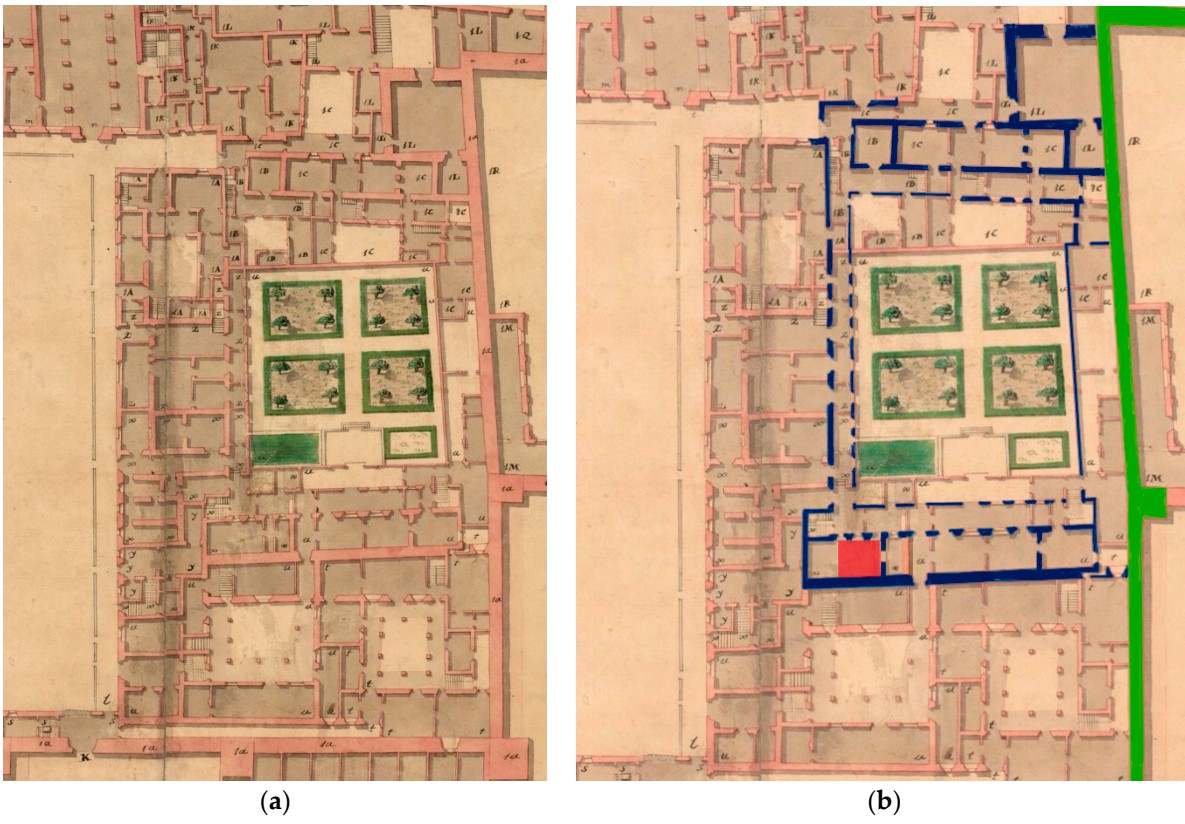

(**a**)                    (**b**)

**Figure 5.** Plan by Sebastián van der Borcht (1759) (Spain, Palace General Archive; Plans, Maps and Drawing Section, no. 4.581). Detail of the western sector of the Patio de Banderas (**a**) and the author's drawing of the walls of the hypothetical Islamic palace and the Hall of Justice (blue) and vault with interlaced arches (red) (**b**).

However, in addition, the existence of a vault of interlaced arches in house no. 3 of the Patio de Banderas (Courtyard of Flags)—made known by José Gestoso y Pérez at the end of the 19th century, and whose construction he attributed to this same period (Gestoso y Pérez [1889] 1984, I. 324–325; [1884] 1914, p. 71)—supports this approach. Regardless of its chronology, it is interesting to note how the aforementioned Sevillian art historian and archaeologist had already suggested its possible link with the Palacio del Yeso (Gypsum Palace), referring to it as a vestige of the primitive palace. Therefore, it is logical to think that in this place there was a palatine area that, due to its dimensions, must have had a great importance in the context of this entire fortified complex—perhaps even thinking that it could be part of the much-acclaimed ʿAbbādī palace mentioned in the Arab sources (González Cavero 2018, pp. 199–217).

At this point, the investigations carried out in recent years in this entire area have allowed us to partially corroborate this approach, in addition to suggesting a new constructive process for the origin of the fortress. According to the results obtained from archaeological interventions and studies carried out previously (Tabales Rodríguez 2013, pp. 99–102; 2020a, pp. 53–113; Tabales Rodríguez and Gurriarán Daza 2021, pp. 1–15; Tabales Rodríguez and Vargas Lorenzo 2014, pp. 16–22; 2017, pp. 198–203; 2020, pp. 21–61; Vargas Lorenzo 2019, pp. 10–13), its walls were built over an 11th century neighborhood outside the walls, eliminating some of the pottery and houses that were part of it (Hernández Souza 2014, pp. 67–70), and many of these constructions survived in its vicinity until the

mid-12th century. It was already in the Almohad period when the southern urban reform of Seville and the expansion of the fortress to the west took place. Currently, the emerging remains that have been preserved from that first moment—and that have traditionally been linked to the *Dār al-Imāra* of the 10th century—correspond to the northern and western walls of the Patio de Banderas (Courtyard of Flags), built from stone ashlars possibly taken from the old Roman wall of the city. This first fortified enclosure had a quadrangular plan, and its axial entrance to the east was flanked by two towers (Tabales Rodríguez 2002a; Gurriarán Daza and Márquez Bueno 2020, pp. 115–49).

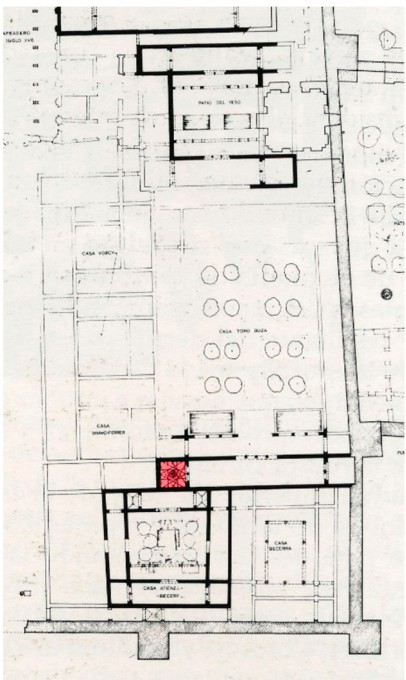

**Figure 6.** Plan of the Reales Alcázares of Seville, according to Manzano Martos (1995a, p. 106). Detail of the western sector of the Patio de Banderas.

It is precisely in its interior where the remains of the aforementioned palace are found—specifically on the western side of the Patio de Banderas (Courtyard of Flags)—where work carried out in recent years by Miguel Ángel Tabales and Cristina Vargas has allowed not only the recovery of some of its constructive and decorative elements[22], but also a hypothetical restitution of how it could have been (Tabales Rodríguez and Vargas Lorenzo 2014, pp. 10–33; 2017, pp. 210–15; Vargas Lorenzo 2019, pp. 1–40; 2020a, pp. 209–58) (Figure 7a,b). We refer to the existence of a tradition caliphal geminate arcade with its corresponding polychromy that opens to what was the western room of the north hall—the northern headwall of which has been preserved, along with some pieces of its ceiling with sunk panels and the foundations of several brick pillars that supported the portico that preceded it. At the same time, a small void with fragments of plasterwork was found in the south wall of the north room, which has been interpreted as one of the three arches that could have opened over the access to the latter to provide the interior with better lighting and ventilation. As for the garden space, it was possible to document the start of a series of arches—currently blinded—in its northwest closing wall and part of a pool that was located in the north front, as well as the partial remains of a central fountain and of the perimeter platforms that delimited it.

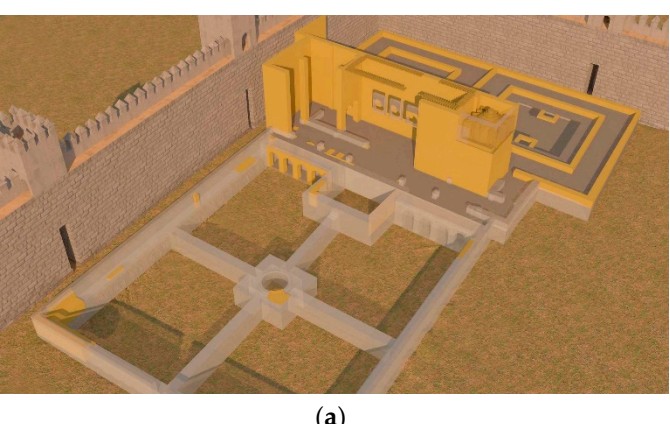 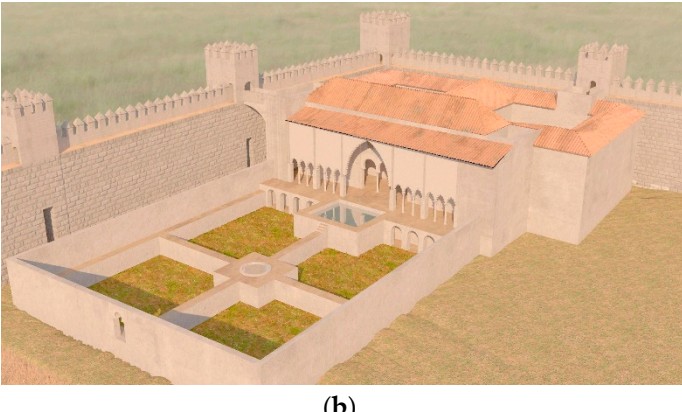

(**a**)                    (**b**)

**Figure 7.** Virtual reconstruction of the excavated remains of the Islamic palace (**a**), and hypothetical restitution (**b**) of Jesús García Carpallo presented by Miguel Ángel Tabales Rodríguez and Cristina Vargas Lorenzo (2019, Figures 16 and 11).

Based on these testimonies, the aforementioned authors have proposed that we are talking about a palace of considerable dimensions, very similar to those of the old palace of the Casa de Contratación of Seville (House of Trade) (Jiménez Hernández 2015, pp. 24–25)—that is, 30.48 m × 42.67 m, with its eastern side being slightly smaller. In turn, it must have had a sunken transept garden whose flowerbeds had a 2.5 m difference in level with respect to the walkways of the palace complex. This landscaped space was equipped with a central fountain at the intersection of its platforms and a pool on the north front, occupying the central axis and flanked by four brick arches—currently blinded—on the north wall of these quadrants, with a purely decorative function. Regarding the latter, and according to the studies carried out previously, no access to the palace has been found thus far from these arcades, whose arches have a thickness of 0.05 m (Vargas Lorenzo 2019, p. 26; 2020a, p. 240).

Although with certain variants and at different scales, this last arrangement can already be seen in other sectors of the Reales Alcázares of Seville (Royal Palace and Fortress of Seville)—as is the case of the Cuarto Real (Royal Bedroom) or the Crucero (Crossing), where a sequence of slightly pointed arches is repeated on four sides to bridge a natural difference in level of almost 5 m, forming a walkable gallery around its entire perimeter; in the second construction stage of the patio of the old Casa de Contratación (House of Trade), with double-blind arches, pointed and tumid, in which the paintings of some doors are still preserved; and in the Patio de las Doncellas of the Palace of Pedro I, in which a succession of a semicircular arches—also blind—intertwine with one another forming, in turn, a series of pointed arches.

As we can interpret from all of these examples, the influence that the different palatine spaces received from one another and from different periods is evident, with their proximity being an essential factor. As in the Cuarto Real (Royal Bedroom) or the Crucero (Crossing), the existing topographic unevenness in the north–south axis of the palace under study (Tabales Rodríguez and Vargas Lorenzo 2017, p. 212; 2020, p. 41; Vargas Lorenzo 2019, p. 18; 2020a, p. 229) could have contributed to the depth of the flowerbeds being so pronounced, arranging that sequence of arches in its walls for a greater articulation of the latter and, thus, configuring practicable galleries around its perimeter. In this sense, it would be logical to think that these arcades were repeated in the fronts of the four spaces destined for the vegetation, although we do not know whether this was truly the case. If not, we should remember that in the first Almohad patio of the old Casa de Contratación in Seville, the walls of the flowerbeds have mural paintings with motifs of multifoil arches with leaves—very similar to those that we find incised in the spandrels of the geminate arches that allow access to the southern hall of the Patio del Yeso (Gypsum Courtyard)—so we could not rule out this possibility either. In any case, the aforementioned authors note

that this palace must have served as inspiration for the Cuarto Real (Royal Bedroom) or the Cuarto del Crucero (Crossing Room).

Continuing with the 11th century palace garden, Antonio Almagro Gorbea (2015a, p. 12) takes the plan of Vermondo Resta as a reference to emphasize the presence of a central fountain at the intersection of the platforms during modern times, which we know—from the archaeological interventions carried out—replaced the one in the Islamic palace, as can be seen in the graphical documentation collected (Vargas Lorenzo 2019, p. 26, Figure 28; 2020a, p. 241, Figure 26). This reality is further evidence of the transformation that this entire area underwent but that, in a certain way, continued to maintain its primitive configuration, which would allow us to derive an idea of the original layout of the garden.

A similar thing happened with the northern pool, although, in the opinion of the aforementioned specialist, it was moved to its northeastern end due to the possible partitioning of the arches of the northern portico in the Christian period and the subsequent construction of a new porticoed gallery (Almagro Gorbea 2015a, pp. 12–13). In this way, this sector was made more habitable, which led to a reduction in the surface area of the courtyard on this side, altering the entire area. This is also shown in the plan of Sebastián van der Borcht. However, according to the aforementioned specialist, it is possible that the absence of the central fountain in the latter was due to some oversight since, in later floors, it still appears. This practice in the Christian period of fitting out more spaces, thereby reducing the garden, can already be seen in the Cuarto Real (Royal Room) or the Crucero (Crossing)—specifically on the south side, with the construction of the Palacio del Caracol (Spiral Palace)—and also in the old Casa de Contratación (House of Trade) of Seville, in both the north and south sectors. The main objective of these interventions after the Christian conquest was to provide these palaces with a greater number of rooms to meet the needs of the new court (Almagro Gorbea 1999, pp. 345–47; 2007b, pp. 193–99).

However, also noteworthy is the absence of another pool on the southern front of the palace under study—as we can see in the *Qaṣr b. Saʿad* of Murcia (1147–1172)—or in the first Almohad courtyard of the old Casa de Contratación (House of Trade), and that the garden of the 17th and 18th centuries did not have one either. It is possible that, as was the case on the northern side and in the aforementioned courtyard of the old Casa de Contratación (House of Trade), the Christian reform had buried it to provide the whole complex with more practicable spaces, following the processes described above, or that the works were the reason for its complete destruction. Even Miguel Ángel Tabales and Cristina Vargas (Tabales Rodríguez and Vargas Lorenzo 2014, p. 16) mention the loss of many of the remains of the 11th century palace as a consequence of its contemporary over-excavation. However, we cannot rule out the possibility that this construction had only one pool, as is the case in the palace of Onda, Castellón (11th century) (Navarro Palazón and Estall I Poles 2011, pp. 74–83; Garrido Carretero 2013, pp. 35–41)—although in this case the pool is located on the southern front—or in the palace of ʿAlī b. Yūsuf, Marrakech (12th century) (Meunié 1952, pp. 27–32), examples used by Bernabé Cabañero Subiza in his study to date the Sevillian palace from the end of the 11th century (Cabañero Subiza 2020, p. 346), where it was also placed by Tabales Rodríguez and Vargas Lorenzo (2014, p. 32). However, it is significant that, in the case of Onda, the pool is located on the south side, taking into account the importance of the northern halls in Andalusian palatial architecture.

Based on the research carried out previously, a portico opened onto the garden on its north side which, together with the main hall that it preceded, was located 1.30 m above the landscaped space, occupying a privileged place with respect to the rest of the complex and, therefore, being considered the noblest area of the palace (Tabales Rodríguez 2020b, p. 262; Almagro Gorbea 2015a, p. 13). This conception of political propaganda taking advantage of the natural promontory on which the latter was erected, in addition to the foundation included, can already be seen in the *Dār al-Mūlk* of *Madīnat al-Zahrāʾ*—the residence of the caliph that was built in the highest area of the palatine city, taking advantage of the topography of the terrain.

The finding of the foundational pillars that might correspond to two of the western columns of the portico has allowed the authors to restore the sequence of its arcade, equipped with a central arch of greater span and fleche, flanked by two sections—each with a triple arcade and separated by brick pillars. Rafael Manzano Martos had already drawn a series of pillars (see Figure 6) that, as far as can be deduced from his reading, do not coincide in plan with the recent proposal of Miguel Ángel Tabales and Cristina Vargas. However, the scarcity of the preserved material evidence makes it difficult to determine its elevation, with specialists using similar models for its restitution, and bearing in mind the intervention carried out in the Almohad period in the old Casa de Contratación (House of Trade) and the Patio del Yeso.

In the same way, the remains of lateral buttresses belonging to the access of the north hall, along with those of a foundational pillar located at a distance of two meters from the western jamb, have led the aforementioned authors to suggest the possible existence of a triple arcade, unlike what we can see in Rafael Manzano's plan, with a possible caliphal-style horseshoe on which three small arches would open, and whose compositional scheme we can appreciate in the northern face of the Patio del Yeso (Figure 8). If this is the case, we find ourselves before a trifora that—as is the case in the southern pavilion of the Taifa palace of the fortress of Málaga, as well as in the palace of al-Ma'mūn of Toledo—recalls the models of *Madīnat al-Zahrā'*, focusing during this time on the Umayyad caliphate of Córdoba, with a clear symbolic intention of legitimization by the Taifa monarchs (Calvo Capilla 2011, pp. 69–92). Also in the Hall of Ambassadors of the Christian palace of Pedro I in the Reales Alcázares of Seville appear three trifora, using columns and capitals from *Madīnat al-Zahrā'* or from some earlier Almohad palace in this same place.

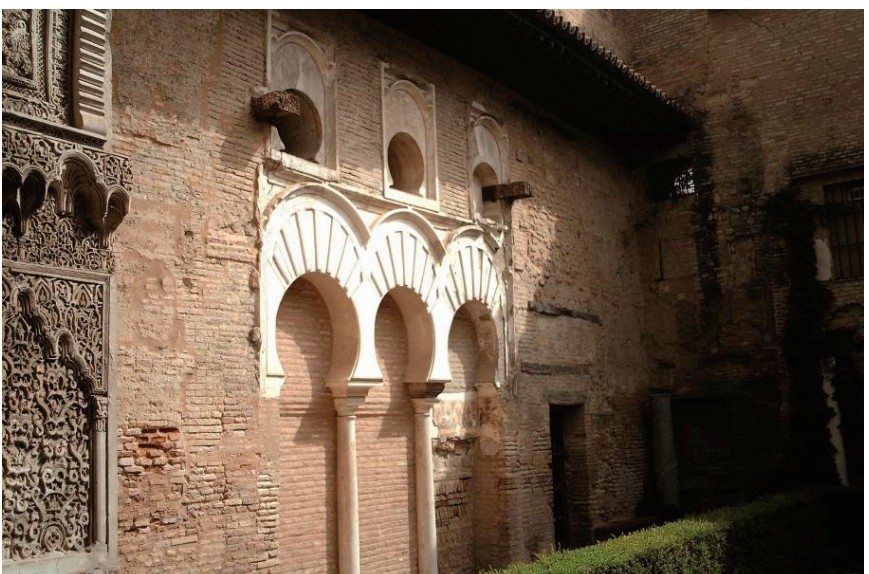

**Figure 8.** Northern face of the Patio del Yeso (Reales Alcázares of Seville). Photo taken by the author.

This idea indisputably underlies the vault of interlaced arches with mocárabes in its keystone that covers what was the eastern room of this hall as a *qubba* (Figure 9). Within it opens a framed geminate arcade (Figure 10)—also of clear caliphal origin—now belonging to house no. 3 of the Patio de Banderas. As we have already mentioned, José Gestoso y Pérez linked the construction of this vault to the Almohad period, taking into account the architectural context in which it was inscribed, followed by Leopoldo Torres Balbás and other more recent researchers[23]. However, given its formal similarities with the vault that covers the space preceding the façade of the *miḥrāb* of the mosque of Tremecén (Algeria), the latter suggested that its construction could have been related to the years of Almoravid presence in the Iberian Peninsula (Torres Balbás 1949, p. 31; 1955, p. 40), towards whose ascription around the late 11th and early 12th century some specialists have positioned

themselves more recently by relating it to the testimonies found in the corresponding north hall of the Sevillian palace (Tabales Rodríguez and Vargas Lorenzo 2014, p. 30; 2017, p. 214; Almagro Gorbea 2015b, pp. 236, 250, 253–54). It has even been suggested that it could be a construction typical of the Taifa period (Cabañero Subiza 2020, pp. 333–34), given its similarities with the disappeared vault of the hot water room of the Aljafería (Torres Balbás 1952, pp. 188–90) and the existence of loop-shaped ornaments of Islamic origin in the keystone of the vaults of the palace of Zaragoza, the palace of Balaguer, and, according to al-ʿUḏrī (1003–1085), in a room of the palace of the Fortress of Almería—although according to Alicia Carrillo Calderero (2009, p. 522), the Almerian geographer placed them in a ceremonial hall of one of the palaces of the governor Muḥammad b. Ṣumādiḥ al-Muʿtasim-.

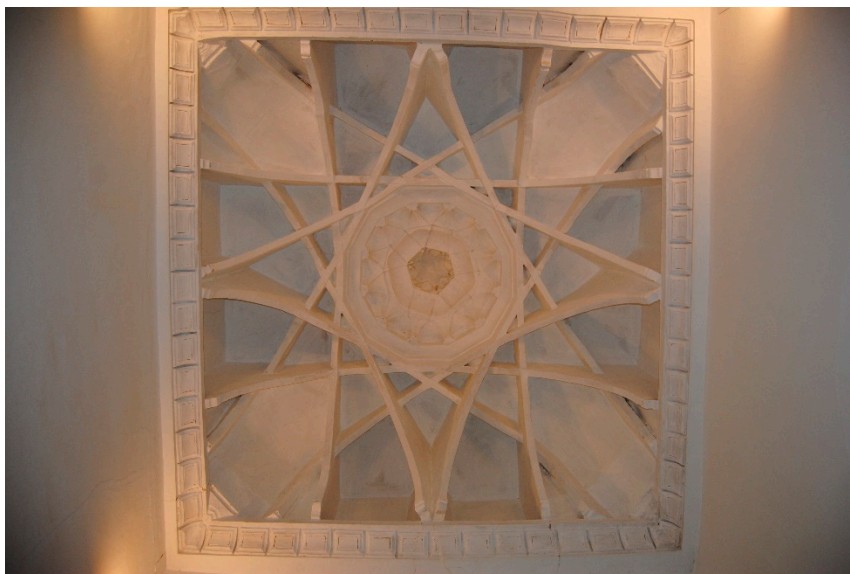

**Figure 9.** Vault with interlaced arches; eastern room of the north hall, house no. 3 of the Patio de Banderas (Reales Alcázares of Seville). Photo taken by the author.

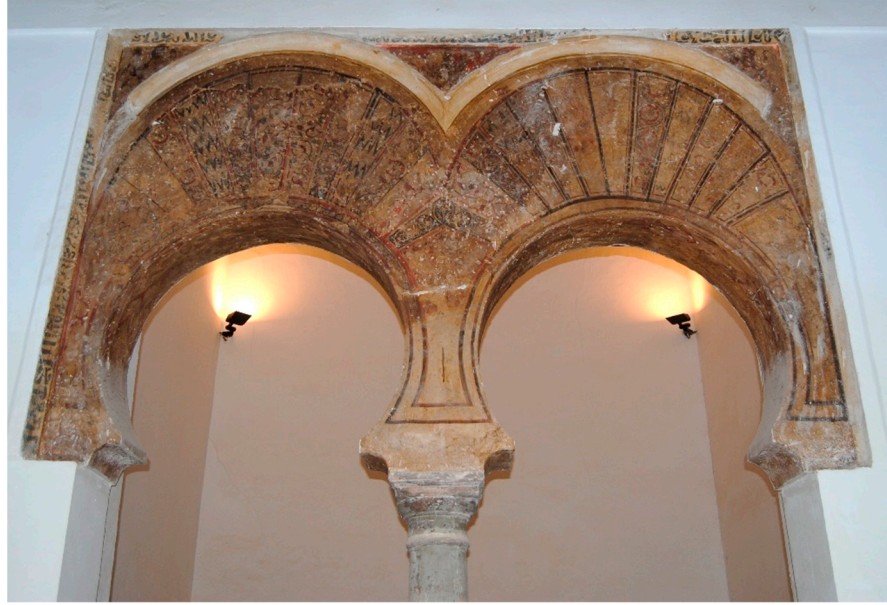

**Figure 10.** Eastern arcade of the north hall, house no. 3 of the Patio de Banderas (Reales Alcázares of Seville). Photo taken by the author.

In any case, the reminiscence of the 10th century is evident, and even the reuse of materials from this period, as can be seen in the shaft and base of the column that make up

the geminated access to the room. In addition, the occupational continuity of the palace can be appreciated in the pictorial decoration of that arcade—based on vegetal, geometric, and epigraphic motifs, and emblems of Castile and León on the spandrels (González Cavero 2018, pp. 213–14)—and in the original alteration of the vault of interlaced arches, with the removal of the latter and the addition of a molding in modern times (Almagro Gorbea 2011, pp. 48–51). All of these aspects make a more precise dating difficult, and a more detailed study would be required in order for us to specify its chronology.

However, one of the most significant contributions of the interventions carried out in the sector of houses no. 7–8 of the Patio de Banderas that could help answer this question was the recovery of the western geminate arcade of the northern hall (Figure 11). Together with the one on its opposite side, they opened onto lateral rooms that flanked a central rectangular space whose original wooden ceiling has been partially preserved. According to the research carried out previously (Vargas Lorenzo 2019, pp. 27–31; 2020a, pp. 244–49), the western arcade has the same decorative motifs that we have just seen, whose analysis has not yet allowed us to define an approximate date for their execution, although it was initially considered that these paintings could date back to the time of this palace's construction—that is, between the late 11th century and the early 12th century (Tabales Rodríguez and Vargas Lorenzo 2014, p. 30; 2017, p. 214).

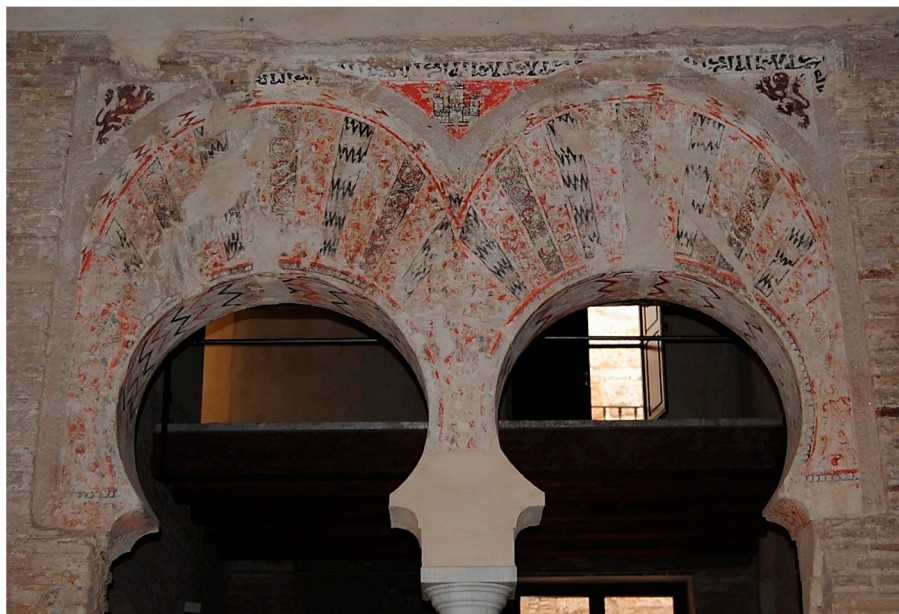

**Figure 11.** Western arcade of the north hall, houses no. 7–8 of the Patio de Banderas (Reales Alcázares of Seville). Photo taken by the author.

Given this lack of precision that we have noted, it has been suggested that the entire pictorial ensemble could correspond to the 14th century, as a result of the reform work carried out after the damage caused by the earthquake of 1356; this would explain not only the lack of precision in its elaboration, but also the homogeneity of the pictorial strata in the words of the specialists[24]. However, we cannot rule out the possibility that—except for the heraldry and the inscription on the framing—it was made during the last years of the reign of al-Muʿtamid, or even that it was replaced during the Christian period.

In this sense, it is striking that in the 14th century tempera painting was used exclusively for the decoration of these arches, rather than plasterwork[25] as in the Patio del Sol (Sun Courtyard), for example (Blasco López and Alejandre Sánchez 2013, pp. 175–82)—a space that belonged to the primitive enclosure of the Royal Alcazars of Seville, and which it seems had a late medieval arcade constructed from this material, pending further progress in the dating of this plasterwork (Baceiredo Rodríguez 2014, pp. 109–30)—or in the current Sala de Justicia (Hall of Justice), in whose place there may have been a similar previous

construction that was renovated in the 14th century (González Cavero 2011, pp. 279–93). We even know that in 1554 the aforementioned "quarto de los yessos" (gypsum room) had plasterwork ornamentation (Gestoso y Pérez [1889] 1984, I. 517–518; Marín Fidalgo 1990, I. 148 and 250), possibly built in the previous century.

While waiting for new research to provide more information about the decoration of these arcades—as well as the existence (or not) of another vault of interlaced arches in the western room of the hall, as has also been suggested—it is significant to note how the motifs of the castle and the rampant lions are further indications that allow us to confirm the occupation of this palace after the conquest of the city by Fernando III (1217–1252) in 1248. To this, we must add the importance of the north hall as a possible protocol or representation area—a reality that would support the use of this palatine complex by the new Christian monarchs, as we have had occasion to comment on the assassination of the Infante don Fadrique.

Another aspect that deserves special attention for a better understanding of the functionality of this palace is the existence of a series of modern houses on its northern and eastern sides, which were identified in the 1990s with the names of their tenants, and to which an Almohad origin was already attributed (Manzano Martos 1999, pp. 65, 72–73). We refer to the "Casa Vorcy", "Casa Bravo-Ferrer", "Casa Atienza-Becerril", and "Casa Becerra"—the latter two bordering the north wall of the hall to which we have just referred[26], as we can see in Sebastián van der Borcht's plan. However, it is precisely the "Casa Atienza-Becerril"—currently house no. 2 of the Patio de Banderas—that preserves some interesting remains that allow us to corroborate that they could have been part of this palace as private residences, used by the monarch or caliph. This house conforms to the prototype of Andalusian domestic architecture, with a central garden surrounded by a platform and a small canal, and with two pools facing one another on the north and south flanks that precede two rectangular halls with rooms; there may also have been other similar pools on the east and west sides (ibid., pp. 72–73; Almagro Gorbea 2015a, p. 13).

Everything seems to indicate that the accesses found in the recovery of the western sector of the north hall should correspond to the communication systems between the latter and the northern homes, although these have not yet been archaeologically documented (Tabales Rodríguez and Vargas Lorenzo 2014, p. 11; Vargas Lorenzo 2019, p. 17; 2020a, p. 226). This gives greater meaning to this area of representation, whose hall the ruler would access directly from his private rooms to officiate any kind of ceremonial act. In this vein, Bernabé Cabañero Subiza (2020, pp. 341–45) establishes a similarity between the building layouts of the mosque of Cordoba, the Aljafería of Zaragoza, and the Sevillian palace, and in the center of the main hall of the latter there may also have been some evocation of the *miḥrāb* of the Cordovan mosque, whose space was reserved for the high dignitaries of the court, as we know was the case in the *Maŷlis al-Šarqī* (Eastern Hall) of *Madīnat al-Zahrā'* (Abad Castro 2009, p. 27, note 27). In the opinion of the aforementioned author, his relatives, members of the court, poets, and musicians, among others, would have been the first to enter the hall through the triple arcade and would be placed—in the same way as in the ḥudī palace—flanking the Taifa monarch, who would enter last, unlike what happened in Zaragoza. Once the ceremony was over, the protocol would be carried out in reverse, with the leader exiting first, followed by the rest of the guests and people linked to the court.

Although we do not have explicit information from the Arab sources about how the royal ceremonies may have taken place in the primitive palace of the Reales Alcázares of Seville (Royal Palace and Fortress of Seville) during the 11th century, in the Aljafería of Zaragoza, both the north portico and the access to the audience hall that precedes it were built with four arches with double columns in the central axis that prevented the monarch from being seen by his attendants; thus, he remained semi-hidden during the act, in keeping with the 'Abbasid and Fatimid tradition (Cabañero Subiza 2012, p. 238; 2020, p. 322). In the case of the Sevillian palace, it seems that this architectural resource did not exist, since the connection between the portico and the hall was made through a triple

arcade in which the central arch opened directly to the space that the ruler was to occupy in the hall.

However, we should not rule out the possibility that this architectural solution was the result of a possible intervention in Almohad times to achieve a greater visual closeness of the caliph to his subjects according to his purposes, returning to the Umayyad tradition (Barceló 1995, pp. 153–75) with this idea of legitimization. Alternatively, it is possible that a simple curtain or veil (*sitr*) hid the Taifa king, as Miquel Barceló (ibid., pp. 155–56) describes for the 'Abbasid and Fatimid caliphates, tracing the origin of this tradition to the East. So, could this be the hall where the caliph Abū Yaʻqūb Yūsuf received the recognition of the Banū Mardanīš in 1172, and even the congratulatory session for the victory against the Christians of Ávila a year later? As we can interpret from the data offered by Ibn Ṣāḥib al-Salā, along with those that we have previously had occasion to gather, the Almohad caliph sat for this type of act "on his lofty and noble throne"—adjectives that refer to the majesty granted to the ruler who uses it (Fierro 2009, pp. 148–49), and to which the term *miḥrāb* is associated as a distinguished space or place (Parada López de Corselas 2013, pp. 113–14), since that "throne" could be as simple as a rug (as Maribel Fierro explains). Hence, the symbolic representation of a *miḥrāb* in the courtrooms should not seem odd, nor should the sources that sometimes allude to the throne with this expression.

Continuing with the descriptions made by the chronicler of Beja on that particular issue, the rest of the representatives of the Almohad state apparatus (*sayyids*, *fuqahā'*, *Ṭālibes*, etc.) were placed next to the caliph in an orderly manner and, once arranged, they proceeded to their presentation and access was given by categories to all those guests waiting at the doors of the palace for their subsequent recognition or congratulation. Then they kissed "his blessed hand", thereby emulating the protocol of the Umayyad Caliphate of Cordoba[27], and whose practice is attributed to the Prophet (Fierro 2009, pp. 136–37). All of this shows us that, at these moments, the first to occupy the throne room were the caliph himself, his relatives, and those close to the court, while the last to enter were those present at the act in question, before participating in a ceremony in which the Almohad caliph showed himself to his subjects without any visual barrier (ibid., p. 143; Marín 2005, II. 451–476).

Regarding the entrance of the ruler to the main hall from his private rooms, and taking into account the hypothetical plans and reconstructions designed for the Islamic palace, it is strange that it was made through the spaces located at its ends, passing previously through the portico to occupy his throne. A direct access would allow the monarch or caliph a certain independence, without the need to mix with the rest of the members of the court who would attend the ceremony, and the hall itself could have contained some access route to the northern houses. This is how we can interpret the existing entrance in the southwest corner of the southern bay of the Atienza-Becerril house that can be seen in the plan of Sebastián van der Borcht (see Figure 5a), which could be a material testimony of the past that has survived to the present day, fossilized and transformed into the door through which we enter today to contemplate the vault of interlaced arches of the eastern hall. Today, this void gives access to the eastern half of what would have been the old north hall of the Islamic palace, and it would be appropriate to wait until a detailed study of it allows us to corroborate its chronology. However, Miguel Ángel Tabales assures us that the origin of this door is from the modern period, and that there must have been an entrance in the eastern hall itself that connected to the adjoining house to the north[28]. One way or another, this idea underlies the configuration of the palatine complex.

However, it is also notable that this palace—as we mentioned when talking about the absence of another pool in the south front of the garden—did not have its corresponding southern corridors (Tabales Rodríguez and Vargas Lorenzo 2014, p. 32). Some specialists have suggested that this circumstance is mainly due to the fact that the construction of this palace coincided with the incorporation of Seville into the Almoravid Empire in 1091 and the consequent exile of the last 'Abbādī sovereign in Agmāt, therefore remaining unfinished (Vargas Lorenzo 2019, p. 17; 2020a, pp. 228–29). Even based on the analysis of

some of the remains, it has been possible to date the beginning of the works of the entire primitive enclosure to the year 1086 (Cabañero Subiza 2020, pp. 305–88). However, the documentary and material testimonies to which we have access, along with the various investigations carried out on the Sevillian palace and fortress, oblige us to review the palace of al-Mu'tamid to help us come closer to understanding it.

### 4. The Palace of Ibn 'Abbād in the Complex of the Reales Alcázares of Seville (Royal Palace and Fortress of Seville): Study and Analysis of Its Chronology and Location

The discovery of the remains of the 11th century palace in the western sector of the Patio de Banderas (Courtyard of Flags) of the fortress of Seville has allowed us to advance our knowledge of the different palatial areas that comprised it, as well as giving us a much closer idea of what could have been the *Qaṣr al-Mubārak*. In addition, the detailed reading of the data offered by the texts on this matter, along with the different studies carried out by specialists from different disciplines, prompts us to approach this reality in an interdisciplinary way, as is increasingly necessary.

Having reached this point, and considering everything that has been revealed so far, we consider it appropriate to answer some questions related to the palace of Ibn 'Abbād. As we have had the opportunity to comment on the data collected in the Arabic written documentation, the most common expression with which the palace appears cited in the context of the 11th century is that of *Qaṣr al-Mubārak*, generalized from the 12th century by the name of "palace of Ibn 'Abbād" and associated by historiography with the figure of al-Mu'tamid of Seville. Thus, we have been able to corroborate it explicitly with 'Abd al-Wāḥid al-Marrākušī who, on the occasion of the assassination of Ibn 'Ammār, identifies the "Qaṣr al-Mubārak" with the "Palace of al-Mu'tamid". Even Ibn Ṣāḥib al-Salā refers in his chronicle to the Sevillian palace as the "palace of Muḥammad Ibn 'Abbād"[29]—the place where Hilāl b. Mardanīš stayed when he went to the Almohad Andalusian capital to recognize the caliph Abū Ya'qūb Yūsuf.

However, we have had more problems regarding its location. Taking into account the information offered by the Arabic sources, as well as the different approaches issued over the years by different authors, everything seems to indicate that it could have been located in the primitive enclosures of the current Reales Alcázares of Seville (Royal Palace and Fortress of Seville), being built in a neighborhood outside the walls that research dates to the late Caliphate and Taifa periods (Tabales Rodríguez 2013, pp. 99–101; 2020a, pp. 58–59; Tabales Rodríguez and Gurriarán Daza 2021, pp. 2–4; Tabales Rodríguez and Vargas Lorenzo 2017, pp. 198–203). To this, we must add how the archaeological interventions carried out by Miguel Ángel Tabales Rodríguez (2020a, pp. 61–64) in the foundation trenches of its walls—traditionally linked to the 10th century *Dār al-Imāra*—have led him to propose a different dating, placing its construction between the mid-11th century and the beginning of the 12th century. With respect to the latter, he proposes two different but close-in-time constructive moments, attributing the works of the foundational palace (Enclosure I) to al-Mu'taḍid, and its extension to the south (Enclosure II) to his son and successor al-Mu'tamid (Tabales Rodríguez 2013, p. 108; 2020a, pp. 57, 110; 2020c, pp. 425–26; Tabales Rodríguez and Gurriarán Daza 2021, pp. 12–13; Tabales Rodríguez and Vargas Lorenzo 2017, pp. 215–18), as already proposed by Magdalena Valor Piechotta (2008, p. 22)[30].

This is something that should not seem strange to us since—as Pilar Lirola Delgado determined from the works of Ibn Ḥayyān (Lirola Delgado 2011, p. 76), followed by other specialists (Valor Piechota and Lafuente Ibáñez 2018, pp. 180, 198)—we know that al-Mu'taḍid built magnificent palaces; the author went so far as to relate the construction of the *Qaṣr al-Mubārak* to the aforementioned monarch, and to which al-Mu'tamid added other dependencies. Even José Guerrero Lovillo (1974, p. 97) already posited that this palace may have existed in the time of al-Mu'taḍid. Apart from the chronology derived from the aforementioned interventions, and although the Cordovan chronicler is not very explicit in this regard, the study of the documentary sources leads us to reflect on this approach and

endorse this hypothesis, as already proposed a few years ago (González Cavero 2013, II. 61 and 73–74)—albeit with certain nuances.

We know that Ibn Zaydūn already made reference in his verses to the existence of this palace prior to the date of his death in 1071, linking it to al-Muʿtamid, as we have been able to verify in the Arabic texts. It seems that the aforementioned vizier and poet entered al-Muʿtaḍid's service in 1049–1050, which is when he must have met a young al-Muʿtamid who, two years after Ibn Zaydūn's arrival at the Sevillian court, was appointed governor of Huelva and Saltés by his father, whereupon he had to leave (Lirola Delgado 2011, pp. 110–13)[31]. It was not until the assassination of his brother and successor Ismāʿīl b. ʿAbbād in 1058–1059 by al-Muʿtaḍid himself that al-Muʿtamid had to return to Seville to be appointed heir in his place (ibid., p. 255). Therefore, we think that it is between the years 1058–1059 and 1071 that the Cordovan vizier and poet may have written the poem to which we have referred; thus, the *Qaṣr al-Mubārak* should have already been built.

We have been able to corroborate this from the study carried out by Auguste Cour, who points out how Ibn Zaydūn—already serving al-Muʿtamid—reminded him through these verses of his duty to maintain and visit his palaces, among which (as mentioned in this composition) was the *Qaṣr al-Mubārak* (Cour 1920, p. 128). Moreover, let us not forget the words of the poet Ibn Ḥamdīs when describing the qualities—probably of al-Muʿtamid—extrapolated to the construction of what we think could be the palace of Ibn ʿAbbād, which lead us to suggest that it was conceived for him from the beginning.

Based on the above, and in agreement with the aforementioned specialists, the construction of the *Qaṣr al-Mubārak* had to begin during the reign of al-Muʿtaḍid, since the two years between al-Muʿtamid's accession to power and the death of Ibn Zaydūn are, in our opinion, too short a timespan for the construction of a palace of such magnitude. Let us also remember that al-Muʿtamid conquered Cordoba just after acceding to the throne in 1069—an event that motivated the return of the poet to his hometown, where he spent his last days (Del Moral Molina 2013, p. 121). To all this, we must add—as we can interpret from Ibn Zaydūn's verses—that the *Qaṣr al-Mubārak* could have been practically completed prior to 1071, as evidenced by the use to which it was already being put by al-Muʿtamid according to the poem describing the author's visit to *al-Ṯurayya*, which would support the hypothesis that the palace was built for this monarch while he was still a prince.

The constructive enterprise of this palace would have continued throughout the years of his reign, extending the primitive enclosure towards the south or adding other dependencies, as has already been pointed out. In fact, it is significant to note the personified dialogue that we can read in the *Risāla* (*Epistle*) of Abū Ŷaʿfar b. Aḥmad between the *Qaṣr al-Mukarram* and the *Qaṣr al-Mubārak*, when the latter states how al-Muʿtamid revived its mention and raised its value:

> "(Al-Mubārak speaks) [ . . . ] When my good star returned and a victorious fate favored my fortune by departing from you to me, and a star rose from you and headed towards me, the lord al-Muʿtamid who revived your old ruins and rejuvenated what was already decrepit as he revived my mention and raised my courage, behold my name was inscribed on the list of the great mansions and recorded in the list of the high palaces, who saw the valleys turn into mountains before I did?" (Lledó Carrascosa 1986, p. 197).

Could he have restored that importance not only by reoccupying it, but also by leaving his material imprint on it? As far as its construction is concerned, we have documentary evidence that it was made of stone ashlars. This is what Abū Ŷaʿfar b. Aḥmad himself states later, when he has the *Qaṣr al-Mubārak* say of itself "At every dawn the visitor surrounds me and after walking around, visits every pillar and every stone" (ibid.). Furthermore, we know from Ibn Ṣāḥib al-Salā that the minaret (*sawmaʿa*) of the Almohad mosque of Seville began to be constructed in the year 1184, using "the stone called "taŷūn al ʿādī", taken from the wall of the palace of Ibn ʿAbbād" (Ibn Ṣāḥib al-Salā 1969, p. 201). For its part, in the translation made by Fátima Roldán Castro from this same fragment, we can read how the master builder Aḥmad b. Baso began "the works and did so (with old

ashlars) of stone transported from the wall of the Palace of Ibn 'Abbād" (Roldán Castro 2002, p. 20)—material that coincides with that which predominates in Enclosures I and II of the Reales Alcázares of Seville (Royal Palace and Fortress of Seville).

We can infer that the palace of al-Mu'tamid was exempt and endowed with a wall of stone ashlars, which were possibly reused (González Cavero 2013, II. 274). This nature can also be seen on the occasions of Alfonso VI's siege of the city of Seville in 1082–1083, the establishment of Almohad troops in the capital in the mid-12th century, and the crucifixion of the rebel Ibn Abī Ŷa'far in 1061–1062—all events that we have had occasion to comment on before, and in which reference was made to the palace of Ibn 'Abbād within the framework of the events that unfolded.

Consequently, and as derived from the research conducted by Miguel Ángel Tabales and Cristina Vargas—who identified the primitive enclosures of the palace as the "core area" of the *Qaṣr al-Mubārak* (Tabales Rodríguez 2013, p. 107; Tabales Rodríguez and Vargas Lorenzo 2017, p. 216)[32]—we believe that the latter was effectively conceived as a construction in and of itself. Even the aforementioned specialists propose that it could have housed in its interior some of the most important palaces that are cited in the documentary sources, all integrated into a single palatine complex. In this sense, Pilar Lirola Delgado already noted that most of these buildings were part of the same complex, even locating the *Qaṣr al-Mubārak* next to the *Qaṣr al-Mukarram* (Lirola Delgado 2011, pp. 164–68), as did Valor Piechota and Lafuente Ibáñez (2018, p. 198).

However, if the *Qaṣr al-Mubārak* is identified with Enclosures I and II of the present-day Reales Alcázares of Seville (Royal Palace and Fortress of Seville), where was the *Dār al-Imāra* located? To this day, this question is still a matter of debate, since we have no more information than that offered by al-Bakrī, according to whom it was already built at the time he wrote his work in 1067–1068. Some authors propose, in the absence of future researchers shedding some more light on the matter—and without yet having material data to support it—that it could be found in the place currently occupied by the Archbishop's Palace (Vargas Lorenzo 2020b, pp. 24–25, 35; Tabales Rodríguez 2020a, p. 58). However, it strikes us that a figure such as al-Bakrī—who also served as a court poet during what may have been the last years of al-Mu'tamid's reign, according to the poems that he wrote in 1085–1086 in the context of the battle of *al-Zallāqa* (García Sanjuán 2002, p. 19; 2008, pp. 43–44)—makes no reference in his geographical work to the palace of Ibn 'Abbād, yet he mentions the city wall of Seville, the Emiral Mosque with its minaret, and the "ancient palace called "Dār al-Imāra" (Al-Bakrī 1982, p. 33). Some authors have even identified it with the *Qaṣr al-Mukarram* and placed it in the vicinity of the *Qaṣr al-Mubārak* (Lirola Delgado 2011, pp. 56, 76; Valor Piechota and Lafuente Ibáñez 2018, p. 198).

This is probably due to the fact that, when al-Bakrī wrote his work, it is likely that he had not yet been in Seville and that the palace of Ibn 'Abbād lacked the relevance that it attained years later. However, aside from this, when he speaks of the *Dār al-Imāra* he does so referring to it as an "ancient palace", which leads us to think that there was already another more recent palatial area of some importance at the time, and that, in keeping with the approaches formulated thus far, it could have been the *Qaṣr al-Mubārak*. Therefore, considering this possibility, everything seems to indicate that al-Bakrī wanted to leave a record of the two most significant constructions that symbolize the political and religious power of every Islamic city, i.e., the palace and the mosque (Maíllo Salgado 1995, pp. 330–33). Hence, in the current state of research, we wonder whether the *Dār al-Imāra* could be a mere expression corresponding to the ancient palace of the Sevillian capital—a construction that, in our opinion, 'Abd al-Raḥmān III (912–961) ordered to be repaired and fortified as a consequence of the damages caused by the internal revolts that arose at the end of the 9th century, and that we have already identified with the *Qaṣr al-Mukarram* of the 11th century (González Cavero 2013, II. 46–52).

Despite the silence of the historiography on the primitive palace of the city during the *fitna* (civil strife) and the reign of the Banū 'Abbād, we know from the Arabic written documentation that it continued to maintain its political–administrative and residential

function (ibid., pp. 41–44, 46–52). This occupational continuity is mainly due to the idea of legitimizing the position of the ruling dynasty[33] and its proximity to Ibn ʿAdabbās's mosque—a reality that should not seem strange to us. However, if we continue reading the *Risāla* (*Epistle*) of Abū Ŷaʿfar b. Aḥmad, the author clearly shows how, prior to the year 1076—the approximate date when he wrote this work to obtain the favor of the sovereign and gain access to the Sevillian poetic court[34]—al-Muʿtamid resided between the *Qaṣr al-Mukarram* and the *Qaṣr al-Mubārak*. Then again, we can also infer this from the aforementioned poem by Ibn Zaydūn, in which the Cordovan poet claimed the visit of the ʿAbbādī monarch to *al-Ṯurayya*—a palatine area belonging to *Qaṣr al-Mubārak*.

Taking into account what we are told by Abū Ŷaʿfar b. Aḥmad, there is no doubt of the importance that the old palace located in the center of the city continued to maintain throughout practically the entire 11th century[35], without forgetting the relevance that the *Qaṣr al-Mubārak* also attained (Lledó Carrascosa 1986, pp. 196–99). Therefore, it is possible that, having been conceived for him while he was still a prince, al-Muʿtamid spent some seasons in the latter while his father held power in the primitive palace or *Qaṣr al-Mukarram*. This constitutes additional evidence of the existence of the *Qaṣr al-Mubārak* prior to the rule of the last ʿAbbādī king, consistent with the theories of aforementioned specialists.

We know from the Dianense author that, at the time of composing his *Risāla* (*Epistle*), al-Muʿtamid was in the *Qaṣr al-Mukarram*. However, it is from the years 1082–1083—on the occasion of the siege organized by Alfonso VI, as mentioned above—that the *Qaṣr al-Mubārak* begins to gain some prominence in the documentary sources, and around which the different events that took place developed, frequently cited by the name of Ibn ʿAbbād's palace. Perhaps the poor state in which the old palace of the city was found and the attachment that al-Muʿtamid had towards it led him to fix his residence and all of the administrative and representation apparatus within it sometime after acceding to the throne.

In fact, we have already seen how the Almohads chose the palace of Ibn ʿAbbād as the place of residence and political headquarters of the Andalusian court after their entry into the Sevillian capital in the mid-12th century, which is evidence that this construction could previously have played the same role, as some authors clearly argue (Valor Piechota and Lafuente Ibáñez 2018, pp. 194–95). However, it is striking that the ʿAbbādī sovereign had to move from his palace, south of the city, to the Mosque of Ibn ʿAdabbas for the Friday noon prayer, which we know was a considerable distance away—that is, inside the city and next to the old palace. Could there have been another mosque in the vicinity of Ibn ʿAbbād's palace that was used by al-Muʿtamid for the celebration of communal prayer (*taŷmī*)?

The written documentation tells us nothing about this. Furthermore, in theory, it might seem strange to us that a city would have two congregational mosques, when this practice was initially forbidden. Moreover, we know from Ibn Ṣāḥib al-Salā that the 9th century mosque continued to serve as a mosque during the first years of the Almohad presence in Seville until, in 1182, the caliph Abū Yaʿqūb Yūsuf ordered by decree that the Friday prayer and the *juṮba* (sermon) be transferred to the new Almohad mosque (Ibn Ṣāḥib al-Salā 1969, pp. 198–99). However, this is not to say that the existence of two mosques of this nature in the same city could not have occurred, as long as there were a series of conditions that justified it (Calero Secall 2000, pp. 125–40).

According to the aforementioned chronicler, the reason for the construction of the new Almohad Mosque was based on how narrow the Emiral mosque had become due to the demographic increase in the population, in addition to the poor state of conservation in which it was found (Ibn Ṣāḥib al-Salā 1969, p. 204). However, in this same context, Ibn Ṣāḥib al-Salā records in his work that, prior to the construction of the new mosque, the Almohads "had chosen in their fortress, in the interior of Seville, a small mosque for their weekday and Friday prayers, but it proved to be narrow when their successors chose it as their residence and as the Almohad envoys increased with troops" (ibid., p. 196). As we can read in this fragment, the author confirms that, in addition to the mosque of Ibn ʿAdabbas, there was another mosque used by the Almohads at the same time.

The presence at that time of two mosques in Seville could have been due to the existing differences in religious practice between the Sevillian population and the Almohads during the first years of their government in the Iberian Peninsula, to which must be added the unstable situation motivated by the confrontations between the Andalusians and the unitary armies that settled in the city towards the middle of the 12th century. In our opinion, the location of this small mosque would likely have been in the space later occupied by the interior palace[36], and in which—according to Julio González—the Almohad mosque was built (*Repartimiento de Sevilla* 1951, I. 527).

If this is the case, that small mosque that the North African dynasty had chosen for daily and community prayers could also have been used by al-Mu'tamid when the *Qaṣr al-Mubārak* became his main residence and representative area of the court, causing him to find himself faced with the need to move the political–religious center of the city to the south, with the maintenance of the Emiral Mosque as the main religious building. That small mosque would later be consolidated with the Almohads, in an area outside the walls that, at the end of the 11th century and beginning of the 12th century, was practically urbanized, as seems to be revealed in the treaty of *ḥisba* (religious police/doctrine) by Ibn 'Abdūn ([1948] 1998) (11th–12th centuries) and also confirmed by archeology, which has identified it with the neighborhood of Ibn Jaldūn, where its growth would have also forced the construction of an additional Sevillian mosque.

Therefore, we think that the importance that the palace of Ibn 'Abbād attained during the reign of al-Mu'tamid led to the beginning of a progressive displacement of the political–religious nucleus to the south—to a fortress that, due to the numerous transformations that its entire space has undergone, has not survived in its entirety to the present day. Let us remember that, when talking about this construction, we must do so referring to the whole fortified complex—that is, to Enclosures I and II, with the former being the origin of the continuous extensions that took place from the second half of the 11th century.

## 5. Final Considerations on the Palace of Ibn 'Abbād

It is possible that this first enclosure of the quadrangular plan was conceived as the residence of the monarch, and where the different ceremonies of the court would have also taken place. This is supported by the houses located to the north and that must have been connected—as we have had occasion to comment—with the palace itself. However, similar to other peninsular examples from the Taifa period—as in the Aljafería Palace in Zaragoza, and possibly also in the current Chapel of Belén in the Convent of Santa Fe in Toledo (Calvo Capilla 2004), or in the Alcazaba of Malaga (Íñiguez Sánchez 2018, p. 338)—the latter must have had a private oratory, which we previously identified with the eastern room of the north hall (González Cavero 2018, p. 146), as suggested by Ricardo Velázquez Bosco (1923, p. 298), i.e., a room that opens to this space through a geminate arcade. Even Antonio Almagro Gorbea previously indicated that it had wooden doors that made it independent of the central hall (Almagro Gorbea 2011, p. 49), whose leaves opened inside the room. Proof of this can be seen in the existence of the pivot holes found on the inner faces of the geminated doors.

The roof, based on vaults of interlaced arches, indicates the importance of this area, to which we must add the small loop-shaped ornaments on the dome that contribute to giving this place a certain importance. Its construction, from our point of view, must have corresponded to the interventions carried out by the Almohads in the primitive enclosures of the palace. Nevertheless, apart from the disputed chronology of its construction, we think that in the 11th century there must have existed a similar vault, whose primitive aspect would be analogous to that of the models mentioned above.

However, the possible presence of a similar room at the opposite end—which would be an anachronism from the point of view of Andalusian architecture—makes us question whether the eastern room really had that religious function that we mentioned. Bernabé Cabañero Subiza suggests that this layout—that is, a central hall flanked by two alcoves topped by vaults of interlaced arches—evokes the space of the *maqṣūra* of the mosque of

Cordoba in the same way that it already occurred in the palace of Zaragoza, albeit more explicitly in the Sevillian capital (Cabañero Subiza 2020, pp. 334, 341–45). Therefore, while waiting for future research to determine the purpose of these two rooms, we think that the use of these types of ceilings corresponds to an Almohad transformation, whose spaces may have been intended for the prayer of the caliph and his family—at least, that is how we should conceive the oriental in the Taifa period—given the strong religious component of this dynasty.

The influence that it received from the Aljafería Palace—together with the decoration of the corbels of the wooden coffered ceiling of the north hall (ibid., pp. 357–84) and of the lateral geminated arcades that, according to some authors, were not finished—led Bernabé Cabañero to date the construction of the primitive palace (as well as the vault of interlaced arches of the eastern room to which we have referred) between 1086 and 1091 (ibid., p. 334)—the latter moment of the incorporation of Seville to the Almoravid Empire, and the reason why this palace was never completed. As for its starting date, we have already seen how, prior to the year 1086, the Arab sources tell us of the existence and habitability of the *Qaṣr al-Mubārak*, so it is strange that it remained unfinished—and even more so in the case of the noble area of the palace.

In addition, the occupational continuity of this entire privileged area of the palace must have been materialized in those later actions that we mentioned, which took place not only in the Almohad period, but also immediately after the Christian conquest of the city in 1248. As can be seen in the motifs of lions and castles in the frieze-worked spandrels of a geminated access to the room of the northern hall (Figures 12 and 13), these motifs are adapted to the architectural framework and to the molding that runs along the extrados of the arches, placed at a later time than the realization of the latter, and that interrupt the development of the epigraphic decoration of the framing. It is even worth noting how the ataurique motifs that we can observe in the voussoirs—whose quartering was made by incisions—are very similar to those that are present in the arcades of the church of San Román in Toledo (ca. 1221), probably made by an artist with a marked Islamic-rooted training (González Cavero 2018, pp. 213–14).

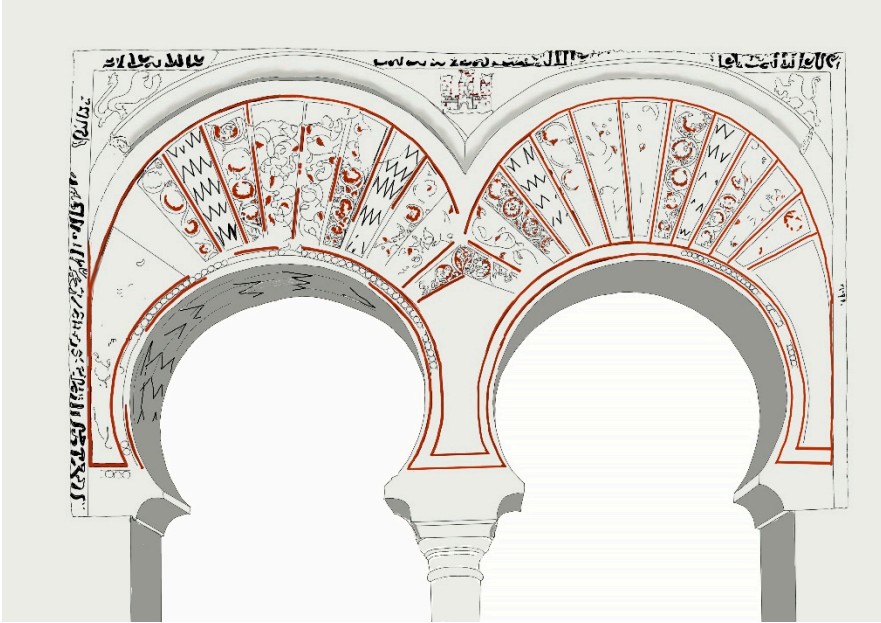

**Figure 12.** Eastern arcade of the north hall, house no. 3 of the Patio de Banderas (Reales Alcázares of Seville). Drawing made by the author.

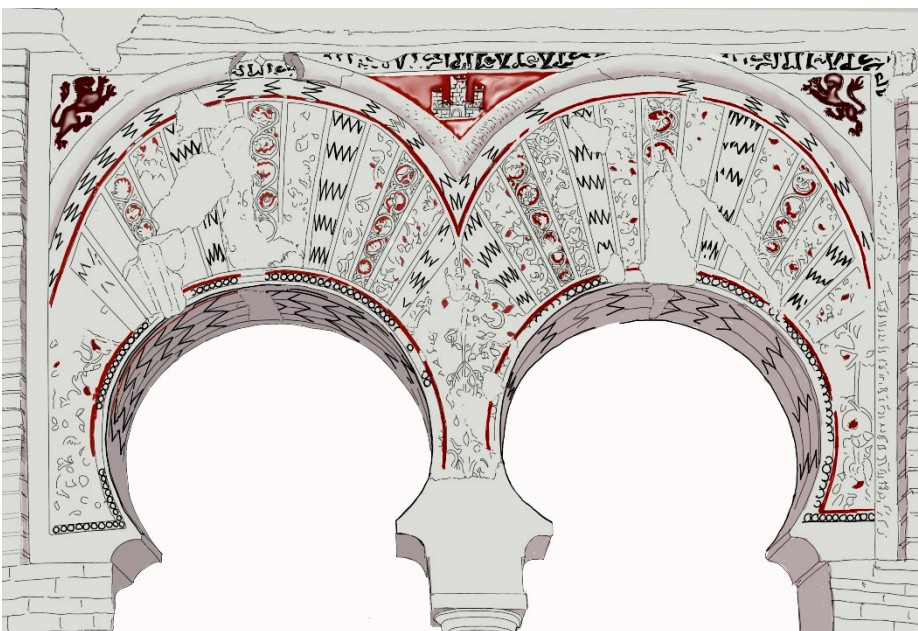

**Figure 13.** Western arcade of the north hall, houses no. 7–8 of the Patio de Banderas (Reales Alcázares of Seville). Drawing made by the author.

However, the absence of plasterwork and the alternation of projecting and depressed voussoirs, to which we must add that other geminated horseshoe arches next to them have practically no decoration—as we can see in the southern hall of the Patio del Yeso or in the old minaret of the Almohad Mosque[37]—leads us to consider the possibility of being faced with an intervention carried out by the Almohads and linked to the reform of the ceilings of their rooms. In fact, it is worth questioning whether this decoration that we see today could correspond to the initial moments of the Christian occupation of the fortress.

Therefore, and based on the above, three possibilities could be considered in terms of the constructive and decorative process: First, that the arches and paintings were made in the 11th century, the latter being subject to later Christian interventions. Second, that the arches date from the 11th century, the paintings from the Almohad period, and that they were also redecorated after the Christian conquest. Third, an ascription of the arches to the 11th century, before being ornamented in a later Christian moment. In relation to this last approach, it is strange that the inscription that runs along the framing—which reads the repeated double expression "joy and prosperity" (*al-yummy wa-l-iqbāl*) in cursive script, first documented in the 12th–13th centuries, just like the one on the opposite arches[38]—was made in the Christian period, when everything seems to indicate that the molding that runs along the extrados of the arches hides part of it, and the motifs of the rampant lions and the castle, after 1248, adapt to the new profile. However, this expression can also be found in Christian contexts, which is why a detailed study of this issue is necessary.

What there is no doubt about is the caliphal origin of the base of the column corresponding to the eastern room that has survived to the present, and which must correspond to that reuse of materials typical of the Almohads in some of their constructions. In addition, Bernabé Cabañero Subiza (2020, pp. 328–29) points out that the column is also originally from the 10th century, when it must have been the capital (replaced in the 16th century), and the supporting elements of the western room eliminated in the 19th century when the access was modified into a flat void (Vargas Lorenzo 2019, p. 21; 2020a, p. 233). If this is the case, this reality would once again corroborate the actions of the Almohads throughout this sector. In addition, the discovery of a marble fragment with the inscription "…de al-Mu'tamid…" after the excavations carried out in the nearby Patio de la Montería (Hunting Courtyard) at the end of the last century (Tabales Rodríguez 2000, p. 29) could support the transformation of the 11th century palace—to which it could have

belonged—by this North African dynasty. Therefore, the latter must have taken place before the beginning of the extension of the primitive enclosure to the west where, over the existing Taifa quarter, the Almohads built their first palace towards the middle of the 12th century (see Figure 2, Enclosure III). We cannot even rule out the possibility that this piece was found decontextualized as a result of the looting that the Almoravids carried out in the palaces of the last 'Abbādī sovereign upon entering Seville in 1091 ('Abd al-Wāḥid al-Marrākušī 1955, p. 112). This could also have entailed the hypothetical later reform that we have commented on, taking into account that the old palace of Ibn 'Abbād was used from the beginning by the Almohads as a political–administrative center.

Rafael Manzano Martos (1995a, p. 115) attributed to the Almohad period some pictorial remains preserved in the triple-blinded arcade that provided access to the northern hall of the Patio del Yeso, without forgetting part of the painted plinth of its southern bay (Valle Fernández and Respaldiza Lama 2000, pp. 56–73), whose manufacture corresponds— as we have just seen in the aforementioned arches—to a clear Caliphate tradition. It is in this space that we have been able to clearly verify the intervention that the Almohads carried out in the whole of the fortress at the architectural and decorative levels, so it should not be surprising that the same thing happened in the palace of the 11th century located in its vicinity, which we believe was all part of the same royal complex, as we will try to argue.

However, focusing on the aforementioned trifora, the investigations carried out on the pictorial decoration that has partially survived in the lower eastern arch and in its corresponding upper void—which we can appreciate on its north face (Figure 14)—have allowed us to delve even deeper into this aspect (Baceiredo Rodríguez et al. 2003, pp. 76–95), proving the existence of three different layers of superimposed polychromy from a detailed analysis of its components. Derived from the latter, it should be noted that although an Almohad ascription cannot be ruled out in terms of the pigments used, the style seems to correspond to the modern period. This practice should not be strange to us, since we have material evidence of how the lower arcades of the Cuarto Real (Royal Bedroom) or the Crucero (Crossing) and the lower walls of the Sala de la Justicia (Hall of Justice) were decorated in this period.

As far as the support itself is concerned—and despite the fact that the central and western arches are the product of a recent reconstruction, along with the small arch above the former—different studies coincide in affirming that this triple arcade of caliphal roots, with alternating projecting and depressed voussoirs, was incorporated into the pre-existing mud walls, and possibly blinded in the 13th–14th centuries (Tabales Rodríguez 2002b, p. 40; Baceiredo Rodríguez et al. 2003, pp. 78–79). According to the stratigraphic analysis carried out by Miguel Ángel Tabales, the north and east walls of the Patio del Yeso are coeval and the oldest among them, dating to the 11th–12th centuries (Tabales Rodríguez 2002b, pp. 40–45). According to the aforementioned specialist, no original voids have been documented in the eastern wall—also a mud wall—extending southward below the eastern geminated arch of the southern Almohad hall. Therefore, it could be a simple enclosure wall. Moreover, given that no previous void has been found, we think that the same thing must have happened in the northern wall, on which the triple arcade dating from the 12th century was later superimposed.

At this point, we know that the north wall was part of a rectangular hall with lateral rooms that have not survived to this day. However, and taking into account the above data, it initially could not be accessed through the Patio del Yeso, ruling out an initial link with the latter. Hence, the functionality of this whole space is still unknown—even more so if we consider that the portico and the southern hall correspond to a later intervention of the Almohad period.

In this sense, we have already seen how the different specialists highlight the absence of a southern hall in the recently discovered 11th century palace, based on several reasons mentioned above. However, in addition to the approaches that we have formulated on this particular matter, and attending to the proportions of this corridor—identified in modern times as "Cuarto de los Yesos"—to its walls—whose prevalence can be dated between the

17th and 19th centuries, just as recorded in graphical documentation—and to its situation with respect to the aforementioned palace, we think that it was none other than the southern hall of the latter (González Cavero 2018, p. 203). In addition, it had to be preceded by a portico and, possibly, by a pool, since it had enough space for one. This can be seen in the hypothetical reconstruction made from the planimetry and the studies available to us, giving us an idea of what this whole area could have been like in the 11th century (Figure 15).

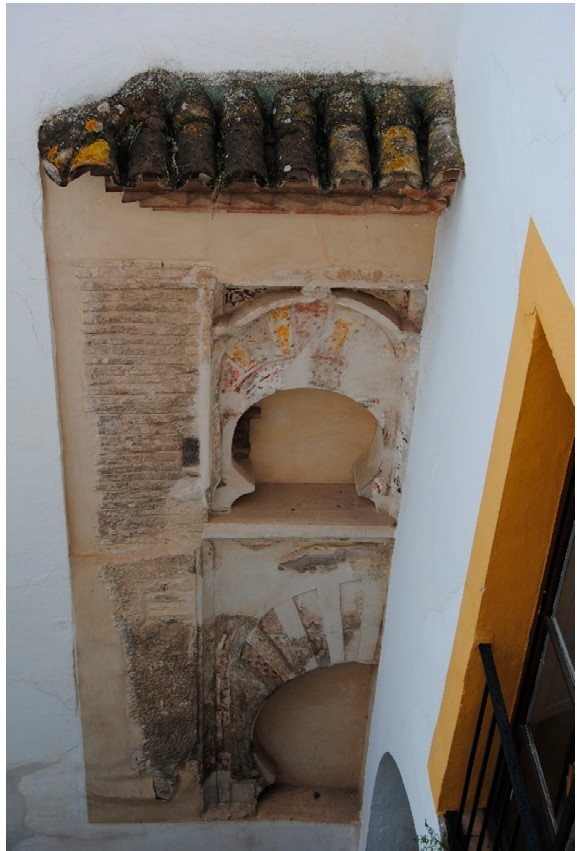

**Figure 14.** North face of the northern wall of the Patio del Yeso (Reales Alcázares of Seville). Photo taken by the author.

It is likely that, if we accept the existence of a *qubba* built in the same century in the place that the Hall of Justice occupies today (Tabales Rodríguez and Vargas Lorenzo 2022, p. 68), there was an access to the latter reserved for the monarch at the western end of the south hall of the 11th century palace, as the plan of Sebastian van der Borcht seems to show clearly. However, a few years ago, we proposed a protocolary function for this primitive *qubba* (González Cavero 2011, pp. 285–88), making us question what was then the purpose of the northern hall of the ʿAbbādī palace. In our opinion, it was all part of the same palatine complex, with the area located north of the Patio del Yeso being the most private and restricted. Its access would have been limited to the high personalities of the court, who could be received by the Taifa king in the main hall.

On the other hand, we think that the *qubba* would have a more public character—that is, reserved for tribute receptions and celebration ceremonies[39]—designating the patio in front of which it stands as a lobby or hall for the embassies or delegations that came to see the monarch. Although we do not know where its entrance could be found—since the eastern and northern walls lacked voids in the 11th century—the access from the outside could have been located somewhere in the southern rectangular room that, according to Miguel Ángel Tabales Rodríguez (2002b, p. 56), could have existed at that time and rested on the southern wall of Enclosure I, which would have been preceded by a narrow portico

(Figure 16). In other words, we are referring to the area where the east wall extended to the south and where, later—after the interventions carried out by the Almohads in the Patio del Yeso—the current southern hall was built.

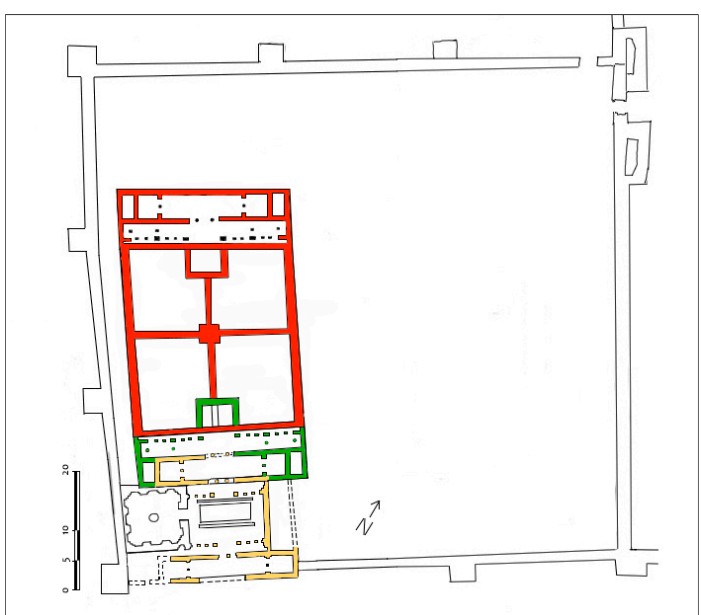

**Figure 15.** Hypothetical composition based on overlapping planes: Miguel Ángel Tabales Rodríguez (red); Antonio Almagro Gorbea (yellow); hypothetical restitution (green). Drawing made by the author.

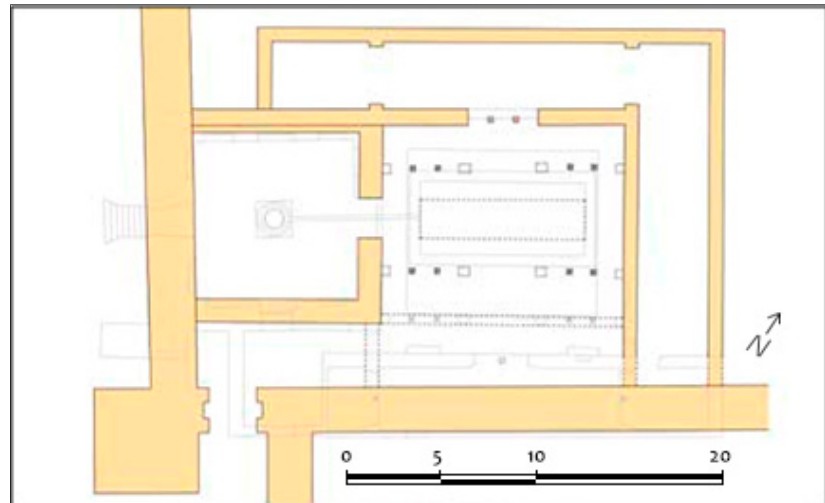

**Figure 16.** Hypothetical plan depicting the origin of the Palacio del Yeso (Tabales Rodríguez and Vargas Lorenzo 2022, Figure 14).

For this, the Almohads had to demolish the southern wall of the primitive enclosure, which allowed them not only to connect the latter with Enclosure II, but also to have a system of latrines in its eastern half[40], while the western half was equipped with a simple room with a bedchamber. As we can see, this southern corridor was not conceived as a residential area as such, nor did it correspond to the model of a rectangular hall with lateral alcoves, which leads us to think that it was a service area and that it did not correspond to the northern hall of the Patio del Yeso. Taking this into account, we find ourselves with yet another clue that would explain, in our opinion, the relationship of the latter with the 11th century palace.

This Almohad transformation to which we are referring also led to the reformulation of this entire space with an artistic language typical of this dynasty, as we can see in the

portico that has survived to the present day. It is possible that the ashlars of the south wall of Enclosure I that were removed during these reforms were reused by the Almohads for the beginning of the construction of the minaret of the mosque, which began in the year 1184 (Ibn Ṣāḥib al-Salā 1969, pp. 200–2). If this is the case, the intervention in the Patio del Yeso must have taken place prior to that date.

Additionally, and with the idea of achieving greater connectivity between the different palatine spaces, it was around that time when the triple arcade of caliphal tradition must have been juxtaposed on the northern wall of the Patio del Yeso. According to traditional historiography, this trifora could have been made in Almohad times or, alternatively, could have been a consequence of its transfer from another nearby place (González Cavero 2018, p. 212), such as the neighboring palace of the 11th century—a possibility that, given the structure of the arches, we think may hold greater weight. One way or another, we can clearly see the Almohads' thoughts of wanting to legitimize their position through stylistic forms corresponding to the moment of greatest splendor of al-Andalus and that, during the Taifa period, were already emulated by the monarchs of the time. The choice of a triple arcade of this nature would clearly correspond to that intention, as would placing it in such a relevant place as the access to the residence of the Almohad caliph from the Patio del Yeso.

Consequently, the southern hall of the 11th century palace would become a transit area, similar to what happened at the western end of the southern bay of the aforementioned courtyard with respect to the Cuarto Real (Royal Bedroom) or del Crucero (Crossing) in Enclosure II (Figure 17). It is precisely in this extension of the primitive palace to the south that we think that the high personalities who came to the Sevillian court could have stayed, as was the case of the Banū Mardanīš. Furthermore, we should not forget that this residential purpose continued to be maintained in the 14th century with María de Padilla. Moreover, its ascription between the 11th and 12th centuries is evidenced in the research carried out on its walls (Tabales Rodríguez 2020a, pp. 79–80). To this we must add that Antonio Almagro Gorbea already noted that there was no absolute dating for the oldest testimonies of this palatine area (Almagro Gorbea 1999, p. 339), with Enclosures I and II being built in a very short span of time, which has led Miguel Ángel Tabales to suggest that we are faced with a construction initiated by al-Mu'taḍid and continued by al-Mu'tamid (Tabales Rodríguez 2020a, p. 110).

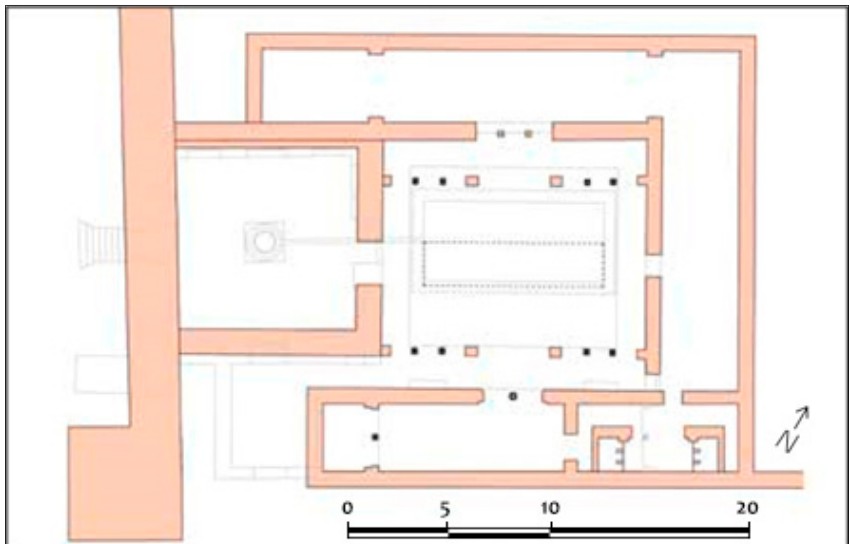

**Figure 17.** Plan of the hypothesis on the development of the Palacio del Yeso at the end of the 11th and beginning of the 12th century (Tabales Rodríguez and Vargas Lorenzo 2022, Figure 14).

## 6. Conclusions

The results derived from the different studies and archaeological interventions carried out in recent years in the enclosures of the Reales Alcázares of Seville (Royal Palace and Fortress of Seville), along with the analysis of the Arabic written documentation, constitute a starting point to advance the knowledge of one of the more relevant palaces of the Sevillian Taifa between the 11th and 13th centuries: the *Qaṣr al-Mubārak*. In this sense, and from what has been exposed in the present study, the splendor that the ʿAbbādī kingdom attained is evidenced in the building works that its monarchs carried out, as recorded in the sources, without forgetting the recent discovery of an 11th century palace of considerable size in the primitive enclosure of the fortress.

Everything seems to indicate that the palace of Ibn ʿAbbād—identified with Enclosures I and II, and to which the texts refer in different ways—was built during the reign of al-Muʿtaḍid of Seville for his successor al-Muʿtamid, who intervened in it by enlarging it as a sign of his power and a reflection of the splendor that his government attained. However, we should not rule out the possibility that this construction was conceived from the beginning as a common project between father and son, in the same way that could have occurred between ʿAbd al-Raḥmān III and al-Ḥakam II (961–976) in the construction of the mosque of Cordoba and *Madīnat al-Zahrāʾ*, or potentially between Yūsuf I (1333–1354) and Muḥammād V (1354–1359/1362–1391) in the palatine city of the Alhambra in Granada.

However, that importance was to be extended to the urban area. In our opinion, it was in the Taifa period that the political–administrative transfer of the city center to the south began, consolidating itself in all those reforms undertaken by the Almohad caliphs from the second half of the 12th century, and in which scenario the palace of Ibn ʿAbbād remained in the memory of this new North African dynasty as a way of legitimizing its position in al-Andalus—a palatine complex whose occupational continuity was felt in the different transformations that were undertaken in later centuries, and whose prevalence in terms of the uses of the different palatine spaces has allowed us to differentiate between the area designated for the monarch and his family, materialized in the 11th century palace and in the housing located to the north of it; the protocolary or semi-public area, in the Patio del Yeso; and the more public area designated for the residence of the representatives of embassies and delegations that came to the Sevillian capital, in the Cuarto Real (Royal Bedroom) or del Crucero (Crossing).

At this point, and as already noted by some authors, it is worth highlighting how the *Qaṣr al-Mubārak* or palace of Ibn ʿAbbād was a palatine complex endowed with different areas where monarchs, caliphs, and members of the court resided and performed their corresponding functions. However, and although we are aware that the new findings and studies carried out have taken a step closer to gaining this knowledge, we consider that further research is still necessary for a better understanding of the "vast and magnificent palace of Ibn ʿAbbād", as the chronicler of the Almohad court Ibn Ṣāḥib al-Salā referred to it.

**Funding:** This research received no external funding.

**Data Availability Statement:** Not applicable.

**Conflicts of Interest:** The author declares no conflict of interest.

## Notes

[1] This work was carried out within the framework of the Excellence Project of the National R&D&I Plan "Chronological systematisation of the Real Alcázar of Seville. Absolute dating and information management through GIS and BIM" (HAR2017-85182-P), funded by the Ministry of Economy, Industry, and Competitiveness.

[2] These and other interventions carried out in the Reales Alcázares of Seville (Royal Palace and Fortress of Seville) can be consulted for free in the work of Tabales Rodríguez (2010, 2016). See also Robador González (2003, pp. 54–56).

[3] Thus, it seems to have been collected from the later work of Al-Ḥimyarī (1938, p. 26 (trans.); 1963, p. 51).

[4] See also Almagro Gorbea (2007a, pp. 165–66) and Valor Piechotta (2008, pp. 65–68), who concur with Miguel Ángel Tabales.

[5] It should be noted that his son, Abū Bakr b. Zaydūn, was al-Mu'tamid's prime minister between 1085 and 1091, after the assassination of Ibn 'Ammār. However, as far as the literary field is concerned, he did not attain the fame of his father (Ben Abdesselem 2001, p. 143).

[6] On this subject, see the translations by Rubiera ([1981] 1988, p. 136) and Pérès (1983, pp. 143–44).

[7] See also Rubiera ([1981] 1988, p. 135) and Pérès (1983, p. 141), as well as Schack ([1944] 2007, p. 190) and Hagerty (2006, p. 193).

[8] "He had, in addition to this, other essential qualities, which are not told, such as courage, modesty and continence, as well as what is related to these noble qualities [ . . . ] Additionally if we name all the good things of The Andalus since it was conquered until now, al-Mu'tamid is one of them or rather the best" ('Abd al-Wāḥid al-Marrākušī 1955, p. 84).

[9] To this, we must add that Ibn Ḥamdīs already mentioned in the first verses of his poetic composition the renovation of the palace to which he refers and that, as we have pointed out, could be the *Qaṣr al-Mubārak*. He begins by saying "Oh what a wonderful abode over which Allah has decided to renew it without it wearing out [ . . . ]" (Rubiera [1981] 1988, p. 136; Pérès 1983, p. 144).

[10] See also the versions made by Dozy ('Abd al-Wāḥid al-Marrākušī [1845] 1881, pp. 87, 90) and Fagnan ('Abd al-Wāḥid al-Marrākušī [1893] 2008, pp. 197, 109). Even later, he again links said palace with al-Mu'tamid of Seville when the Taifa sovereign left his palace to confront the Almoravid armies in the face of the siege and imminent entry of North African troops into the city in 1091 ('Abd al-Wāḥid al-Marrākušī 1955, p. 111; [1845] 1881, pp. 98–99; [1893] 2008, p. 120).

[11] See the translation by Ramírez del Río (2004, p. 239). In this context, and taking into account what other authors report about this event, the fact that the assassination of Ibn 'Ammār took place in the Sevillian palace is confirmed by Ibn Jallikān himself, who refers to the latter generically as "palace of Seville" (Ibn Jallikān 1868, III. 127).

[12] This siege was motivated as a response to the murder of the Jewish treasurer of Alfonso VI by al-Mu'tamid when he went to Seville on the occasion of the payment of parias (Lirola Delgado 2011, pp. 180–81).

[13] For his part, Ibn Abī Zar' (1310–1320) also makes reference to this event, adding that the siege of Seville lasted three days—as Lirola also reports (Lirola Delgado 2011, p. 181)—but without mentioning the place set by Alfonso VI for the meeting of his troops, nor the palace of Ibn 'Abbād (Ibn Abī Zar' 1860, p. 202; 1964, I. 277).

[14] See the translation by Viguera Molins (1998, p. 19; 2005, II. 727–728), as well as Ramírez del Río and Valor Piechotta (1999, p. 174), and the Arabic edition (Ibn 'Iḏārī 1985, pp. 35–39).

[15] In the Arabic edition of 'Abd al-Hādī al-Tāzī we can read "Qaṣr Muḥammad, the prince, the noble" (Ibn Ṣāḥib al-Salā 1964, pp. 472–73).

[16] Ambrosio Huici Miranda refers again on this occasion to the Palace of al-Mu'tamid by the name of "castle of Ibn 'Abbād" (Ibn Ṣāḥib al-Salā 1969, p. 195). For his part, 'Abd al-Hādī al-Tāzī does so using the expression "place of Ibn 'Abbād" (Ibn Ṣāḥib al-Salā 1964, p. 473). Nevertheless, everything seems to indicate that we are dealing with the same palatine area.

[17] This translation has been made by different specialists in different publications (Viguera Molins 1998, pp. 19–20; 2005, II. 729; Ramírez del Río and Valor Piechotta 1999, p. 174).

[18] See also Roldán Castro (2002, p. 19).

[19] At the same time, the author identifies the caliph's residence with the *Qaṣr al-Mukarram* of the Taifa period, locating it at the present site of the Reales Alcázares of Seville (Royal Palace and Fortress of Seville). For her part, Pilar Lirola Delgado (2011, pp. 56, 76) relates the *Qaṣr al-Mukarram* with the *Dār al-Imāra* or ancient palace/alcazar (*qaṣr al-qadīm*), located very close to the *Qaṣr al-Mubārak*, and raising the possibility that it was the residence of the caliph Hišām II, which could also have been the administrative center.

[20] "This was when Muhammad b. Sa'd b. Mardanīs died, he hastened his son Hilāl to present himself to the Caliph, after settling Abū Hafs in Murcia. It was his arrival with all his brothers and his father's companions, caids and the greatest ones of his soldiers, at the end of Sa'bān of this year (end of April 1172). They went out to meet him the Sayyid Abū Zakarīya and his brother Ibrahīm, brother of the Caliph, with a group of Almohads; and in their company he entered the Caliph's hall, near the evening prayer on the day of his arrival" (Ibn 'Iḏārī 1963, p. 441).

[21] It should not be forgotten that the so-called "quarto del Sol" (Sun's room), located in the current house no. 11 of the Patio de Banderas, became the home of the lieutenant governor from the 18th century, which once again evidences the importance of the entire sector of the primitive palace and its subsequent expansion to the south. In addition, there is evidence that this space was occupied and renovated in the 14th century, as shown by the preserved materials (Bañasco Ibáñez et al. 2018, pp. 69–90).

[22] We would like to thank Miguel Ángel Tabales for his kindness and willingness when receiving us, and for the interesting on-site explanations about the entire recovery process of this area of the Reales Alcázares complex in Seville (Royal Palace and Fortress of Seville), as well as for his comments and observations in this regard.

[23] The chronology proposed by José Gestoso y Pérez was adopted, among others, by Rafael Manzano Martos (1995a, p. 117) and Antonio Almagro Gorbea (2011, pp. 45–53; 2013, pp. 89–90), the latter pointing out a certain parallelism with the vault of the upper chamber of the minaret of the Kutubiyya Mosque in Marrakech.

[24] In addition to Cristina Vargas's observations, Bernabé Cabañero affirms that the decorative motifs, which are similar to those found in the vestibule of the palace of Tordesillas, were made in the 14th century and, therefore, there was no previous ornamentation (Cabañero Subiza 2020, pp. 328–29).

[25]    We would like to thank Professor Concepción Abad Castro for her observations in this regard. In addition, it is worth mentioning the existence of the remains of plasterwork belonging to the void over the central arch that allowed communication between the portico and the north hall. However, we do not have more data that would allow us to contextualize these pieces.

[26]    Even the palace to which this room corresponds was built on the site later occupied by what Rafael Manzano called the "Casa Toro-Buiza" (Manzano Martos 1999, pp. 65, 72–73).

[27]    This same protocol can be seen, for example, at the court of al-Ḥakam II, whose description is reported by Ibn Ḥayyān (987–1076) from the chronicler 'Īsā b. Aḥmad al-Rāzī (d. 980) in his *Muqtabis* (Ibn Ḥayyān 1967, pp. 45–46, 70–71, 225–26).

[28]    We would like to thank Miguel Angel Tabales for all of the comments provided in this regard.

[29]    Let us recall that the full name of al-Mu'tamid was Abū l-Qāsim Muḥammad b. 'Abbād b. Muḥammad b. Ismā'īl b. Muḥammad b. Ismā'īl b. Qurayš b. 'Abbād b. 'Amr b. Aslam b. 'Amr b. 'Iṯāf b. Na'īm al-Lajmī, his father's being al-Mu'taḍid Abū 'Amr 'Abbād b. Muḥammad b. Ismā'īl b. Muḥammad b. Ismā'īl b. Ismā'īl b. Qurayš b. 'Abbād b. 'Amr b. Aslam b. 'Amr b. 'Iṯāf b. Na'īm al-Lajmī.

[30]    For his part, Alejandro Jiménez Hernández (2020, pp. 414–15) confirms that the average dating for the foundational citadel coincides with the reign of al-Mu'taḍid, albeit with the chronology of Enclosure II more towards the beginning of the 12th century.

[31]    On the importance Ibn Zaydūn attained at the 'Abbādī court, the study by Auguste Cour (Cour 1920, pp. 97–132) is of great interest.

[32]    However, Cristina Vargas goes so far as to suggest in a more recent publication that the *Qaṣr al-Mubārak* could have been built inside Enclosure II—specifically, in the area later occupied by the Palacio del Caracol (Palace of the Snail) (Vargas Lorenzo 2020b, pp. 33, 37). For their part, Valor Piechotta and Lafuente Ibáñez identify the first enclosure of the palace with the palace of Ibn 'Abbād, and even suggest later that it could be located inside it (Valor Piechota and Lafuente Ibáñez 2018, pp. 197–98).

[33]    Moreover, let us not forget how the supposed Hišām al-Mu'ayyad (Hišām II) was proclaimed in 1035 by Abū l-Qāsim Muḥammad (1023–1042) with that very intention (*Crónica anónima de los reyes de taifas* 1991, p. 73).

[34]    The importance of Seville in this cultural field can be consulted in the work of Salah Khalis (1966).

[35]    It is significant to note how, in the *Risāla* (*Epistle*) of Abū Ŷa'far b. Aḥmad, the *Qaṣr al-Mubārak* addresses the *Qaṣr al-Mukarram* with these words: "[ . . . ] you are the base of the caliphate, the stability of the leadership and the seat of dynasties and the center of uninterrupted reigns [ . . . ] the mass was tranquilized by the administration of the judge of justice and the justice of a brave and virtuous 'Abbād [ . . . ]" (Lledó Carrascosa 1986, pp. 196–97). As we can read, the antiquity of the *Qaṣr al-Mukarram* as a political–administrative seat is clear, and it may even be referring to the qāḍī Abū l-Qāsim Muḥammad b. Ismā'il b. 'Abbād (1023–1042) on the occasion of the events that occurred in the city of Seville during the fitna and the emergence of the Sevillian Taifa.

[36]    Let us remember that, in addition to the Kutubiyya Mosque, the Almohads had another mosque in the *qaṣba* of Marrakech. Furthermore, we know that there were other examples, such as in the Alhambra of Granada, Almeria, Guadix, Ceuta, or Ronda (Calero Secall 2000, pp. 136–37).

[37]    Unlike these examples, the geminate arches of the eastern and western rooms of the north hall are tiled, as we see in some examples of the 10th century, and also in the northern wall of the Patio del Yeso.

[38]    We would like to thank María Antonia Núñez for her remarks regarding this issue. See also Martínez Núñez (2014, pp. 151–53).

[39]    Recently, a religious purpose—or even an administrative one—has been suggested for this area (Tabales Rodríguez and Vargas Lorenzo 2022, pp. 68 and 70).

[40]    It is precisely in this place where Rafael Manzano Martos located these latrines that, at first, he dated as an Almohad work (Ballesteros Beretta 1978, p. XV), before later advancing their chronology to the 11th century and relating them to a domestic space (Manzano Martos 1995a, p. 111). This residential character is already highlighted by Antonio Almagro Gorbea (2015a, pp. 9–11), arranging a series of rooms around the Patio del Yeso. For his part, Miguel Angel Tabales links their construction to the Almohad period (Tabales Rodríguez 2002b, p. 56)—an approach that would make sense if we take into account that, in the 11th century, the southern wall of Enclosure I ran through the same area.

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
