# Peer review of "Seville, a Cultured and Influential Court: The Palace of Ibn ‘Abbād"

_arts, 2022_

Round 1

Reviewer 1 Report

This is an extremely important article with several original insights into new discoveries in Seville, published in English for the first time. However, it is let down significantly by the poor quality of the translation. The English is comprehensible, but rather awkward. In the first place I recommend using DeepL to create a preliminary translation, and then finding a native English speaker to correct it, with sight of the Spanish original. This article offers an important argument, but the translation makes it very difficult to follow, even for someone familiar with the Alcazares.

Line 42 refers to 'Enclosures I and II' in Fig 1. Presumably this refers to 'Recintos I and II' in Fig. 1, but it would be helpful to clarify this, and to indicate the location of the palaces of Pedro I and Alfonso X so that readers can orient themselves. 

The translation beginning on line 92 would be more helpful if it showed the original Arabic, rather than a translation into Spanish. 

It would be helpful to include some lines on the reliability of al-Marrākušī. Had he been to Seville, and/or what sources did he draw on? The same applies to all the Arabic sources discussed: Ibn Ṣāḥib al-Salā is rightly considered to be a fairly reliable source, but what about the others? It is not enough just to gather these sources - there needs to be a clearer sense of which are the most important. The very recent publication by Juan Martos Quesada - https://medievalistas.es/1-historiografia/ - may be helpful for this.

Is it assumed that there was no palace near the original Umayyad congregational mosque (now San Salvador), as in Cordoba and elsewhere?

line 513: what is the relevance of the reference to the 'palace of iron' in the chronicle of Pedro I? Why could it not refer to the so-called Gothic palace, or the Almohad palaces? Presumably the 'cuarto del caracol' refers to spiral staircases, like those in the Gothic palace? Fig 5b is very striking, but it would be better to use a higher quality drawing below, like that used for 5a. In both 5b and in Fig. 6 it would also be helpful to indicate the site of the dome with interlacing arches (in Casa no. 3). 

723 - it is surely worth mentioning that there three triphora also appear in so-called Salon de los Embajadores in the palace of Pedro I, and that these use spoliated columns and capitals from ?Madinat al-Zahra? or ?an earlier palace on the site?

846 - connections with Zaragoza are also strengthened as the Aljaferia was similarly known as Qasr al-Mubarak

It is rather confusing that both sections 1 and 4 refer to the Arabic texts, with some repetition. Could these be clustered together, perhaps after the archaeological evidence is summarized? 

1115 - does Ibn Ṣāḥib al-Salā refer to the mosque in the Alcazar as a congregational mosque or simply a neighbourhood mosque? It would be unusual only to have two of the former.  The evidence for the existence of a large mosque on this site prior to the Almohad congregational mosque seems very slim - if it was replaced by the Almohad mosque then surely Ibn Ṣāḥib al-Salā would describe this in his very detailed account? Ibn Ṣāḥib al-Salā also records that Ahmad b. Baso, architect of the Almohad mosque, was also responsible for 'raising domes' in Cordoba. 

1218 -In English 'Unitarians' usually refers to a branch of the Protestant church. Some other alternative to Almohads should be used. 

1228 - 'al-yumn wa-l-iqbal' is not strange in a Christian context, but very common. It is found, for example, in San Roman in Toledo. Stylistically the lions and castles look like they belong soon after 1248, though it is hard to be certain about this. Under Pedro I one would expect to find the arms of the Orden de la Banda as well. All this is significant because it may suggest that Alfonso X made his palace here, and not in the Gothic palace, which - contrary to the claims of most scholars - cannot be definitely linked to him. 

1235 - who is the aforementioned specialist? Better to give his name again as the reader must search back a long way. 

Reviewer 2 Report

This paper constitutes an excellent state of the arts about the research on Reales Alcázares de Sevilla, particularly on the palace of Ibn 'Abbād. It compiles both written and archaeological information to build solid hypotheses about how and where this palace can be identified and when it was built, also considering the immediate urban surroundings. It is well referenced and takes into consideration the most recent archaeological and architectural works carried out by M. A. Tabales and his team. All in all, this paper updates the archaeological knowledge about this very complex enclosure. I would suggest minor improvements related to the figures that illustrate the text. Some of them should incorporate a scale and a north (for instance, figs. 15, 16 or 17). Some others should be enlarged because they have important details that are difficult to perceive as they are right now (for example, figs. 1 to 6). 
Apart from this, English needs to be reviewed and improved.

Reviewer 3 Report

Very interesting paper that supposes a considerable advance in the knowledge of the Reales Alcázares of Seville, and especially of its origins in the XI century. It is an article that combines different sources in a coherent and solid way, methodologically impeccable and with a graphics curated.

Reviewer 4 Report

The study is mainly dealing with the palace of Ibn 'Abbād and its possible location in the current site known as the Reales Alcázares from Seville. Data was taken from archaeological interventions carried out in several sectors of this palatine complex, studies that have dealt with this issue and rigorous analysis of the documentary sources the authors had. The results wanted to highlight the importance this palace had between the 11th and 13th centuries and its relationship with the rest of the official and residential spaces that make up the Alcazar of Seville nowadays.

The article is based on a series of studies and very extensive documentation. I would like to congratulate the authors for the detailing and careful study of the smallest traces that provided information for outlining the main objective of the article. The interdisciplinary way of the approach made by the authors offered a very good interpretation of the text and studies used as references.

Following the narrative course, I notice problems in the clear definition of the hypotheses and the methodology used, but probably the multiple sources of analysis and the involvement of the authors emphasized a different way of organizing the content of the article.

I recommend the authors to introduce hand sketches/ CAD drawings throughout the narrative to complete the information acquired and the personal interpretation.
